# Nanoparticle delivery of microRNA-146a regulates mechanotransduction in lung macrophages and mitigates injury during mechanical ventilation

Christopher M. Bobba[1,2,3,6], Qinqin Fei[1,2,4,6], Vasudha Shukla[2,3], Hyunwook Lee [1,3], Pragi Patel[1,3], Rachel K. Putman[5], Carleen Spitzer[1], MuChun Tsai [1], Mark D. Wewers[1,3], Robert J. Lee[4], John W. Christman[1,3], Megan N. Ballinger[1,3], Samir N. Ghadiali [1,2,3,6✉] & Joshua A. Englert [1,2,3,6✉]

Mechanical ventilation generates injurious forces that exacerbate lung injury. These forces disrupt lung barrier integrity, trigger proinflammatory mediator release, and differentially regulate genes and non-coding oligonucleotides including microRNAs. In this study, we identify miR-146a as a mechanosensitive microRNA in alveolar macrophages that has therapeutic potential to mitigate lung injury during mechanical ventilation. We use humanized in-vitro systems, mouse models, and biospecimens from patients to elucidate the expression dynamics of miR-146a needed to decrease lung injury during mechanical ventilation. We find that the endogenous increase in miR-146a following injurious ventilation is not sufficient to prevent lung injury. However, when miR-146a is highly overexpressed using a nanoparticle delivery platform it is sufficient to prevent injury. These data indicate that the endogenous increase in microRNA-146a during mechanical ventilation is a compensatory response that partially limits injury and that nanoparticle delivery of miR-146a is an effective strategy for mitigating lung injury during mechanical ventilation.

[1] Division of Pulmonary, Critical Care, and Sleep Medicine, Department of Internal Medicine, The Ohio State University Wexner Medical Center, 473 West 12th Avenue, Columbus, OH 43210, USA. [2] Department of Biomedical Engineering, The Ohio State University, 140 West 19th Avenue, Columbus, OH 43210, USA. [3] The Davis Heart and Lung Research Institute, The Ohio State University Wexner Medical Center, 473 West 12th Avenue, Columbus, OH 43210, USA. [4] Division of Pharmaceutics and Pharmacology, College of Pharmacy, The Ohio State University, 500 West 12th Avenue, Columbus, OH 43210, USA. [5] Division of Pulmonary and Critical Care Medicine, Department of Internal Medicine, Brigham and Women's Hospital, 75 Francis Street, Boston, MA 02115, USA. [6]These authors contributed equally: Christopher M. Bobba, Qinqin Fei, Samir N. Ghadiali, Joshua A. Englert. ✉email: ghadiali.1@osu.edu; joshua.englert@osumc.edu

The acute respiratory distress syndrome (ARDS) occurs in patients with pneumonia, sepsis, trauma, or other pulmonary insults including COVID-19 and is characterized by loss of alveolar-capillary barrier integrity and the development of pulmonary edema. Clinically, this manifests as impaired oxygenation, reduced lung compliance, and bilateral radiographic infiltrates on chest x-ray[1,2]. Although supportive care with mechanical ventilation (MV) is the standard of care for ARDS patients, the physical forces generated during MV can exacerbate lung dysfunction through a phenomenon known as ventilator-induced lung injury (VILI)[3,4]. VILI can also occur in mechanically ventilated patients without ARDS or those undergoing MV during surgery[5,6]. Injury from MV is induced through a variety of mechanisms including overdistention from high tidal volumes (volutrauma, tensile force), increased airway pressure (barotrauma, compressive force), or inadequate positive end expiratory pressure (PEEP) leading to alveolar collapse and reopening (atelectrauma, shear force). These forces directly damage the alveolar-capillary barrier, triggering the release of mediators and recruiting inflammatory cells that further damage the barrier (i.e. biotrauma)[3,4]. A more thorough understanding of the molecular mechanisms by which cells respond to mechanical forces may reveal exploitable biologic pathways that could be targeted via novel therapeutics to treat or prevent VILI. This is of utmost clinical importance given that the current management of patients with ARDS involves using MV for life-support and treating the underlying cause of ARDS with the hope that lung injury will resolve spontaneously. Unfortunately, multiple failed clinical trials[7] indicate that completely eliminating the injurious mechanical forces during ventilation is not possible and there are no pharmacologic therapies to treat or prevent the development of lung injury in these mechanically ventilated patients. Instead of eliminating the injurious mechanical forces, here we propose an alternative approach where endogenous compensatory responses to these mechanical forces are used to develop therapeutically relevant gene delivery approaches that can mitigate the development of lung injury during MV.

The alveolar-capillary barrier is an extremely thin structure consisting of epithelial and endothelial cell layers. To date, most studies investigating the injurious effects of MV have focused on how these structural cells sense and respond to mechanical forces and have shown that injurious mechanical forces during volutrauma, barotrauma, and atelectrauma cause plasma membrane disruption, cell detachment, and activation of pro-inflammatory signaling[8–11]. In addition to structural cells, alveolar macrophages (AMs), a resident immune cell population in the lung, are known to contribute to the pathogenesis of lung injury including VILI[12–14]. AMs reside within the alveolar airspaces in close proximity to epithelial cells where they are exposed to mechanical forces during MV. Previous work has shown that macrophages are mechanosensitive and respond to cyclic stretching by increasing pro-inflammatory cytokine production[15]. However, the mechanisms by which AMs release cytokines during MV is not known.

MicroRNAs (miRs) are small non-coding RNA molecules that act as negative post-transcriptional regulators and play an important role in disease pathogenesis[16]. Altered miR expression plays an important role in the pathogenesis of a variety of lung diseases including cystic fibrosis[17], pulmonary fibrosis[18], sarcoidosis[19], and lung cancer[20]. Disease-specific regulation makes miRs viable biomarkers, and specific miRs have been identified in acute lung injury[21]. A prior study in mice found that miRs related to innate immunity and inflammation (i.e. miR-146a and miR-155) were significantly upregulated in lung tissue in response to injurious MV[22]. However, little is known about how mechanical forces during ventilation alter miR expression in human subjects and in specific cell types. Yehya et al.[23] reported increased miR-466 expression in rat alveolar epithelial cells subjected to cyclic stretching and our lab previously demonstrated increased miR-146a expression in primary human lung epithelial cells exposed to cyclic transmural pressure[24]. We have also reported that over-expression of miR-146a reduced pressure-induced inflammation in human epithelial cells by targeting key elements of the toll-like receptor signaling pathway (IRAK1 and TRAF6)[24]. However, it is not known if miR-146a is dysregulated in human subjects undergoing MV, if miR-146a is a mechanosensitive miR in AMs, or if modulating miR-146a expression in AMs in vitro or in vivo can be used to dampen lung injury and inflammation caused by MV.

In this study, we tested the hypothesis that miR-146a regulates the mechanotransduction processes in AMs that lead to VILI using both in vitro and in vivo models. We specifically show that injurious forces during MV increase miR-146a levels in human and murine AMs and that miR-146a null mice have increased lung injury following MV. We demonstrate that the endogenous increase in miR-146a in AMs following injurious ventilation is not sufficient to prevent lung injury. However, when nanoparticles are used to increase miR-146a to supraphysiologic levels, lung injury during MV can be prevented. Although the endogenous increase in miR-146a during MV is an insufficient compensatory response that partially limits lung injury, supraphysiological increases in miR-146a levels using a nanoparticle delivery technique is an effective strategy for mitigating lung injury during MV.

## Results

**VILI upregulates miR-146a in human AMs**. To determine the mechanosensitivity of primary AMs, these cells were isolated from human donor lungs and subjected to 16 h of oscillatory pressure at an air–liquid interface as an in vitro model of barotrauma during VILI[24]. Pressure-induced changes in pro-inflammatory cytokine secretion and miR-146a expression were measured (Fig. 1a, b and Supplementary Fig. 1). We focused on IL8 release in response to oscillatory pressure given that increased levels of IL8 in the circulation and bronchoalveolar lavage (BAL) are associated with worse outcomes in ARDS patients[25,26]. Although baseline IL8 secretion varied by donor, exposure to barotrauma increased IL8 secretion across all donors (Fig. 1a). There was also a consistent increase in miR-146a expression following oscillatory pressure in all donors (Fig. 1b). We also investigated the macrophage response to oscillatory pressure using a cell line where PMA-differentiated THP1 cells were exposed to oscillatory pressure as described above. These cells also exhibited increased IL8 secretion when exposed to oscillatory pressure compared to control cells not exposed to pressure (Fig. 1c). Similar to primary human AMs, miR-146a expression in pressurized THP1 cells increased approximately twofold (Fig. 1d). To determine if force induced miR-146a is a clinically relevant phenomenon, miR-146a levels were measured in BAL cells obtained from a cohort of patients who underwent a clinically indicated bronchoscopy. This cohort included spontaneously breathing patients, mechanically ventilated patients without ARDS, and mechanically ventilated patients with ARDS. The baseline characteristics of the subjects are shown in Table 1. There were no significant differences between groups in demographics. Patients undergoing MV had higher rates of shock and were therefore more likely to require vasopressor support. There were no significant differences in tidal volume, PEEP, or duration of MV prior to bronchoscopy in mechanically ventilated patients with or without ARDS. As expected, patients with ARDS on MV required a higher amount of supplemental oxygen (FiO$_2$) compared to mechanically ventilated patients without ARDS. BAL

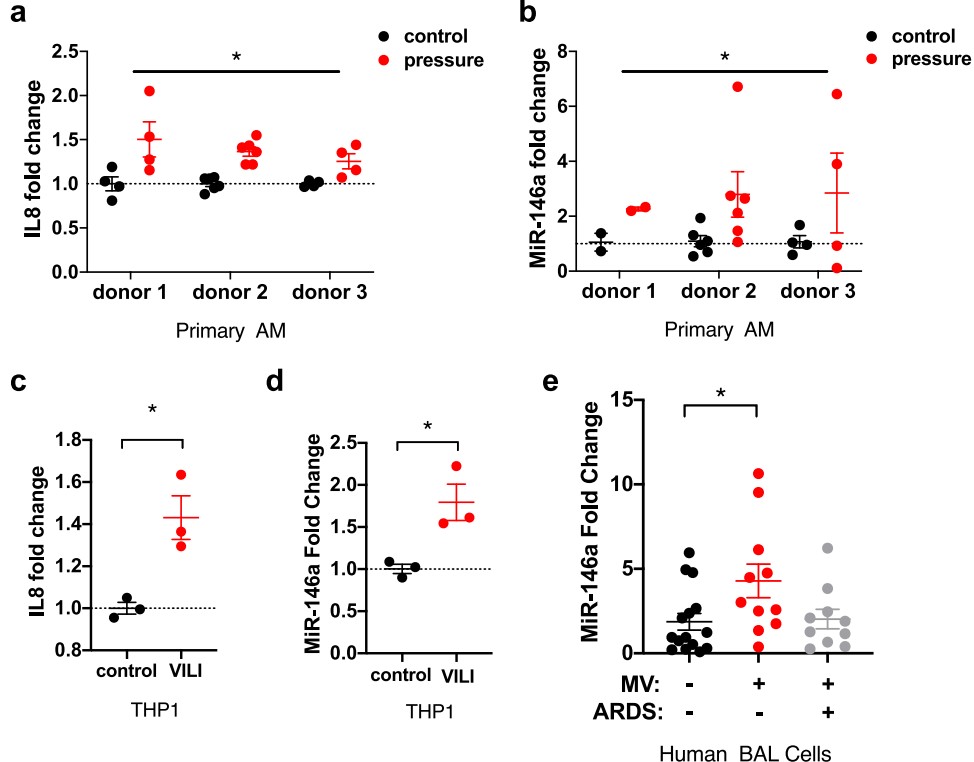

**Fig. 1 miR-146a is upregulated in macrophages during ventilator-induced lung injury. a** Fold change in IL8 secretion in alveolar macrophages (AMs) from three donor lungs subjected to 16 h of oscillatory pressure at an air–liquid interface, compared to unpressurized controls. *indicates that pressure injury is a statistically significant factor, $p < 0.0001$, via two-way ANOVA on normally distributed data, $n = 4$ for donors 1 and 3, $n = 6$ for donor 2. **b** Relative miR-146a expression in AMs subjected to pressure, calculated by $\Delta\Delta$Ct method, normalized to unpressurized controls. *Indicates that pressure injury is a statistically significant factor, $p = 0.0475$, via two-way ANOVA on normally distributed data, $n = 2$ for donor 1, $n = 6$ for donor 2, $n = 4$ for donor 3. **c** Fold change in IL8 secretion from THP1 cells subjected to 16 h of oscillatory pressure at an air–liquid interface (VILI), compared to unpressurized controls. Data normally distributed, $p = 0.0160$ via two-tailed Student's $t$-test, $n = 3$ wells per group. **d** Relative miR-146a expression in THP1s subjected to oscillatory pressure, calculated by $\Delta\Delta$Ct method, normalized to unpressurized controls. Data normally distributed, $p = 0.0239$ via two-tailed Student's $t$-test, $n = 3$ wells per group. **e** miR-146a expression in RNA extracted from BAL cells in a cohort of patients undergoing a clinically indicated bronchoscopy. Relative expression determined by $\Delta\Delta$Ct method, normalized to no mechanical ventilation (MV). Data log-normally distributed, $p = 0.0411$ via ANOVA with post-hoc Tukey on $\log_2$ (fold change) comparing MV no ARDS and no MV no ARDS groups; $n = 15$ for no MV group, $n = 11$ for MV no ARDS, $n = 10$ for MV with ARDS. Data are presented as mean ± SEM.

fluid contained a mixed inflammatory infiltrate consisting primarily of macrophages and neutrophils and there were no differences between groups in the total number of BAL cells or differential cell counts (Table 1). In comparison to spontaneously breathing patients, miR-146a expression was increased in mechanically ventilated patients without ARDS (Fig. 1e) indicating that physical forces during MV upregulate this miR. Interestingly, miR-146a levels in mechanically ventilated patients with ARDS were similar to spontaneous breathing (SB) controls. Given that IL8 is a neutrophil chemokine and that BAL fluid contained a mixture of AMs and neutrophils, we explored whether neutrophils could be contributing to the increase in miR-146a following VILI. Peripheral blood neutrophils (polymorphonuclear cells (PMNs)) were isolated from healthy donors and cultured at an air–liquid interface (Supplementary Fig. 2a, b). PMNs were subjected to in vitro barotrauma and similar to AMs, PMNs were mechanosensitive and released IL8 in response to barotrauma (Supplementary Fig. 2c). However, there was not a consistent increase in miR-146a levels with only one of three donors having increased miR-146a levels following in vitro barotrauma (Supplementary Fig. 2d). Furthermore, miR-146a expression levels in neutrophils following barotrauma were nearly 100-fold lower compared to AMs (Supplementary Fig. 2e). These data indicate that both primary human AMs and PMNs are

mechanosensitive and release IL8 in response to in vitro barotrauma. However, only human alveolar macrophages exhibit a consistent increase in miR-146a levels in response to injurious forces generated during MV.

**Injurious MV in mice increases AM miR-146a levels.** To investigate if miR-146a expression increases in response to injurious MV, wild-type (WT) mice were subjected to MV with high tidal volumes (12 ml kg⁻¹) without PEEP for 4 h to induce lung injury[27,28]. Cells were isolated from BAL fluid and the majority of cells from mechanically ventilated and SB control animals were macrophages (Fig. 2a and Supplementary Fig. 3a–c). miR-146a expression was assessed in BAL cells and increased about tenfold following 4 h of VILI (Fig. 2b). This was consistent with our observation in BAL cells from mechanically ventilated patients (Fig. 1e). BAL concentrations of IL6 (Fig. 2c) and CXCL1/KC (Fig. 2d), the murine homolog of IL8, were increased in ventilated mice compared to SB controls. BAL total protein, a surrogate for alveolar-capillary barrier permeability, was also increased following ventilation (Fig. 2e). The combination of increased inflammation and barrier dysfunction resulted in increased lung elastance (i.e. stiffness, Fig. 2f) and impaired oxygenation following ventilation (Fig. 2g). In summary,

**Table 1 Patient characteristics and group analysis from Intensive Care Unit cohort[a].**

| | No MV (N = 15) | MV (N = 11) | MV + ARDS (N = 10) | Overall (p-value) | No MV vs MV (p-value) | No MV vs MV + ARDS (p-value) | MV vs MV + ARDS (p-value) |
|---|---|---|---|---|---|---|---|
| Age (years) mean ± SD | 56.4 ± 15.4 | 53.5 ± 17.5 | 59.2 ± 9.8 | 0.69 | 0.66 | 0.62 | 0.38 |
| Sex, no. female (%) | 6 (40) | 5 (45) | 5 (50) | 0.89 | 1.0 | 0.70 | 1.0 |
| Race, no. white (%) | 10 (71) | 10 (91) | 9 (90) | 0.36 | 0.34 | 0.36 | 1.0 |
| White blood cell count, median (IQR) | 6.6 (2.0, 9.9) | 9.4 (6.9, 16.2) | 16.9 (11.9, 22.9) | 0.04 | 0.069 | 0.0060 | 0.11 |
| Lactate, mean ± SD | 1.33 ± 0.51 | 2.14 ± 1.23 | 2.04 ± 1.28 | 0.42 | 0.19 | 0.28 | 0.87 |
| Vasopressors, no. yes (%) | 0 | 7 (64) | 3 (30) | 0.0007 | 0.0005 | 0.052 | 0.20 |
| SIRS median (IQR) | 1 (0, 3) | 2 (2, 3) | 4 (3, 4) | 0.0075 | 0.19 | 0.0054 | 0.043 |
| Systolic blood pressure <90, no. yes (%) | 0 | 9 (82) | 5 (50) | <0.0001 | <0.0001 | 0.0047 | 0.18 |
| BAL cell count (median, IQR) | 15 (7, 28) | 15 (3, 38) | 24 (9, 35) | 0.82 | 0.92 | 0.50 | 0.72 |
| Total ($10^4$ cells) | | | | 0.98 | 0.95 | 0.90 | 0.82 |
| Macrophages (%) | 29 (4, 81) | 33 (5, 70) | 43 (16, 73) | 0.52 | 0.56 | 0.64 | 0.23 |
| Neutrophils (%) | 43 (15, 79) | 62 (18, 94) | 36 (15, 72) | 0.74 | 0.44 | 0.80 | 0.64 |
| Lymphocytes (%) | 1 (0, 3) | 1 (0, 2) | 2 (0, 4) | 0.16 | 0.37 | 0.058 | 0.33 |
| Eosinophils (%) | 0 (0, 0) | 0 (0, 0) | 0 (0, 1) | | | | |
| $PaO_2{:}FiO_2$[b] median (IQR) | – | 239 (190, 265) | 126 (103, 209) | – | – | – | 0.11 |
| Tidal volume (ml kg$^{-1}$) mean ± SD | – | 7.3 ± 3.5 | 6.7 ± 0.9 | – | – | – | 0.59 |
| Time on ventilator before bronchoscopy (hours) median (IQR) | – | 35 (14, 110) | 13 (6, 67) | – | – | – | 0.22 |
| PEEP (cmH$_2$O) median (IQR) | – | 6 (6, 8) | 8 (6, 8) | – | – | – | 0.62 |
| $FiO_2$ (%) mean ± SD | – | 47 ± 9 | 68 ± 25 | – | – | – | 0.034 |

*MV* mechanical ventilation, *ARDS* Acute Respiratory Distress Syndrome, *SD* standard deviation, *IQR* interquartile range, *SIRS* systemic inflammatory response syndrome, *BAL* bronchial alveolar lavage.
[a]Missing data—lactate: 10 in the no MV group, 1 in the MV group, and 3 in the MV + ARDS group; bronchial alveolar lavage cell counts: 2 in the no MV group, 1 in the MV group.
[b]$PaO_2{:}FiO_2$ is the ratio of the partial pressure of oxygen in arterial blood to the fraction of inspired oxygen.

injurious forces during MV upregulate miR-146a expression about tenfold in AMs during murine VILI.

**miR-146a$^{-/-}$ mice have increased injury during MV.** To assess whether miR-146a plays a protective role in modulating VILI, miR-146a knockout (KO) and WT litter-mate control mice were subjected to injurious MV for 4 h and compared to SB control mice. There was no difference between BAL cytokine or protein concentrations in miR-146a KO mice and WT mice prior to ventilation (SB in Fig. 3a–c). Following ventilation, KO mice exhibited significantly increased BAL IL6 and CXCL1/KC levels (Fig. 3a, b) and elevated total protein concentrations (Fig. 3c) compared to ventilated WT mice. Differential cell counting revealed that global deletion of miR-146a did not significantly alter the recruitment of inflammatory cells following injurious ventilation (Fig. 3d). KO mice also demonstrated a significant increase in lung elastance (Fig. 3e) and decrease in oxygenation (Fig. 3f) following ventilation compared to WT controls. These data demonstrate that loss of miR-146a results in increased proinflammatory cytokine secretion, disruption of the alveolar-capillary barrier, and impaired lung function compared to WT animals. Together, the data in Figs. 2 and 3 indicate that miR-146a is an important negative regulator of lung injury during MV. However, the endogenous increase in miR-146a in WT mice in response to MV is not sufficient to abrogate VILI.

**Endogenous miR-146a levels are insufficient to mitigate VILI.** A variety of cell types may be responsible for increased miR-146a

expression during injurious MV in vivo. To determine the extent to which AMs contribute to VILI, mice were treated with liposomal clodronate to deplete AMs or vehicle control prior to ventilation[29]. This method led to a 50–60% reduction in the number of total cells and AMs (Supplementary Fig. 4a, b) in the BAL and a decrease in neutrophils (Supplementary Fig. 4c). Consistent with a prior study in rats[13], clodronate depletion of AMs decreased lung inflammation (Supplementary Fig. 4d, e) and barrier disruption (Supplementary Fig. 4f), and improved lung function (Supplementary Fig. 4g) during MV. miR-146a levels in BAL cells were not statistically different between vehicle control and liposomal clodronate groups, which is likely due in part, to variability between clodronate and vehicle-treated animals (Supplementary Fig. 4h). However, we did observe a statistically significant correlation between miR-146a levels and the number of AMs in the BAL suggesting that AMs are an important source of miR-146a (Supplementary Fig. 4i). These data also suggest that AMs play a dual role in lung injury during MV. AMs are mechanosensitive and their responses to injurious forces contribute to the development of VILI (Supplementary Fig. 4d–f) and AMs also possess a compensatory pathway to mitigate lung injury by upregulating the expression of miR-146a during VILI (Figs. 1b, d, e and 2b).

To determine the capacity of AM-derived miR-146a to mitigate lung injury, we adoptively transferred bone-marrow-derived macrophages (BMDMs) from WT and miR-146a KO mice into WT and KO recipient mice 6–8 h prior to injurious MV. In the first experiment, we adoptively transferred WT and KO BMDMs into KO mice. The transfer of WT cells into miR-146a KO mice

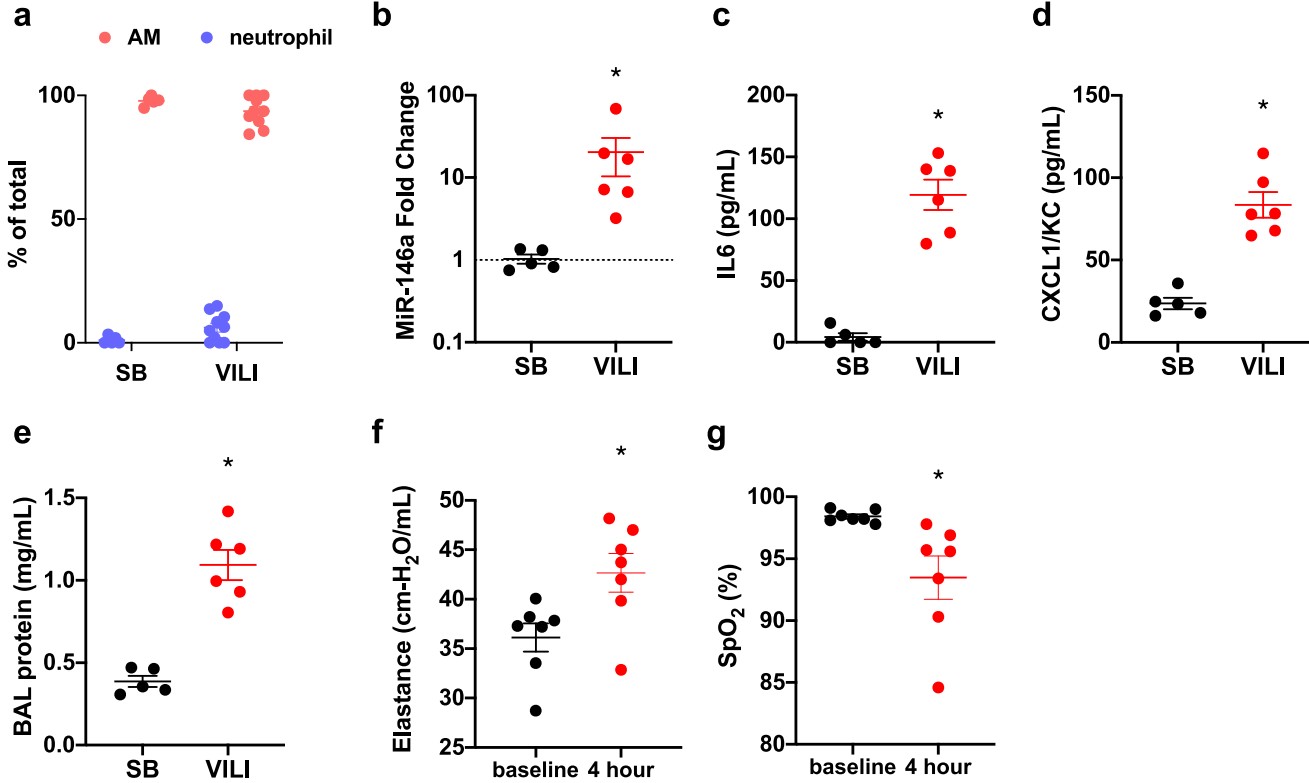

**Fig. 2 Injurious mechanical ventilation in mice increases miR-146a levels in alveolar macrophages. a** Bronchoalveolar lavage (BAL) differential cell counts in spontaneously breathing (SB) and ventilator-induced lung injury (VILI) groups, $n = 5$ for SB, $n = 10$ for VILI groups. **b** miR-146a expression from RNA extracted from BAL cells following 4 h injurious ventilation (VILI) or from SB controls. Relative expression determined with ΔΔCt method, normalized to SB. Data log-normally distributed, $p = 0.0007$ via two-tailed Student's $t$-test on $\log_2$ (fold change), $n = 5$ for SB, $n = 6$ for VILI groups. **c** IL6 concentrations from SB and VILI groups; $p = 0.0043$ via Mann–Whitney test, $n = 5$ for SB, $n = 6$ for VILI groups. **d** KC/CXCL1 concentration from SB and VILI groups. Data normally distributed, $p < 0.0001$ via two-tailed Student's $t$-test, $n = 5$ for SB, $n = 6$ for VILI groups. **e** BAL protein concentration from SB and VILI groups. Data normally distributed, $p < 0.0001$ via two-tailed Student's $t$-test, $n = 5$ for SB, $n = 6$ for VILI groups. **f** Lung tissue elastance measurements at initiation of ventilation (baseline) and at the conclusion of 4 h ventilation. Data normally distributed, $p = 0.0194$ via two-tailed Students $t$-test, $n = 7$ per group. **g** Blood oxygen saturation (SpO$_2$) was measured by pulse oximetry at initiation and conclusion of ventilation. Data normally distributed, $p = 0.0158$ via two-tailed Student's $t$-test, $n = 7$ per group. Data are presented as mean ± SEM.

did not lead to any change in lung injury. The miR-146a KO mice that received WT BMDMs had similar levels of BAL inflammatory cytokines and inflammatory cells (Fig. 4a, b and Supplementary Fig. 5, left panels), lung function (Fig. 4c, left panel; Fig. 4d), and barrier permeability (Fig. 4e, left panel) after injurious MV compared to KO mice that received KO BMDMs. miR-146a levels in BAL cell pellets were modestly increased in KO mice that received WT BMDMs (Fig. 4f, left panel). These data demonstrate that although there was a statistically significant increase in miR-146a expression after adoptively transferring WT BMDMs into miR-146a KO mice, the level of miR-146a expression was not sufficient to mitigate VILI. In a separate set of experiments, WT and miR-146a KO BMDMs were adoptively transferred into WT recipient mice prior to injurious MV. Interestingly, WT mice that received miR-146a KO BMDMs had significantly higher levels of BAL IL6, KC, and neutrophils (Fig. 4a, b, Supplementary Fig. 5, right panels) compared to WT mice that received WT BMDMs. Despite these differences in cytokine levels and neutrophils, there were no significant differences in lung elastance (Fig. 4c, right panel), oxygenation (Fig. 4d, right panel), or barrier permeability (Fig. 4e, right panel). Importantly, miR-146a levels were significantly decreased in the WT mice that received KO BMDMs when compared to miR-146a levels in WT mice that received WT BMDMs (Fig. 4f, right panel). Data from these adoptive transfer experiments indicate that the lack of endogenous miR-146a in AMs leads to enhanced

inflammation during VILI while the administration of WT BMDMs is not sufficient to rescue the increased lung injury seen in miR-146a KO mice. This is likely due to the fact that the increase in miR-146a levels after adoptive transfer of WT BMDMs into KO mice was modest (approximately fivefold, Fig. 4f, left panel) and similar to the endogenous increase observed in WT mice during ventilation (Fig. 2b). We posit that modest increases in miR-146a seen during MV or during adoptive transfer of WT BMDMs into KO mice is insufficient to mitigate lung injury and that alternative methods that can dramatically upregulate miR-146a expression are needed to mitigate VILI.

**Nanoparticle miR-146a delivery decreases VILI in vitro.** To determine if miR-146a overexpression can be used to regulate force-induced IL8 production, lipofectamine or a lipid nanoparticle (LNP) delivery system was used to increase miR-146a levels in primary human AMs (Fig. 5a). To fabricate miR-146a-loaded LNPs, we first generated empty LNPs using ethanol dilution and sonication. Polyplexes of pre-miR-146a and polyethylenimine (PEI) were then mixed and sonicated with empty LNPs to create miR-146a-loaded LNPs. This protocol yielded solid LNPs (Supplementary Fig. 6a) with an average diameter of $160.8 \pm 7.3$ nm (mean ± SEM, $n = 2$, Supplementary Fig. 6b, c), an average polydispersity index (PDI) of $0.393 \pm 0.022$

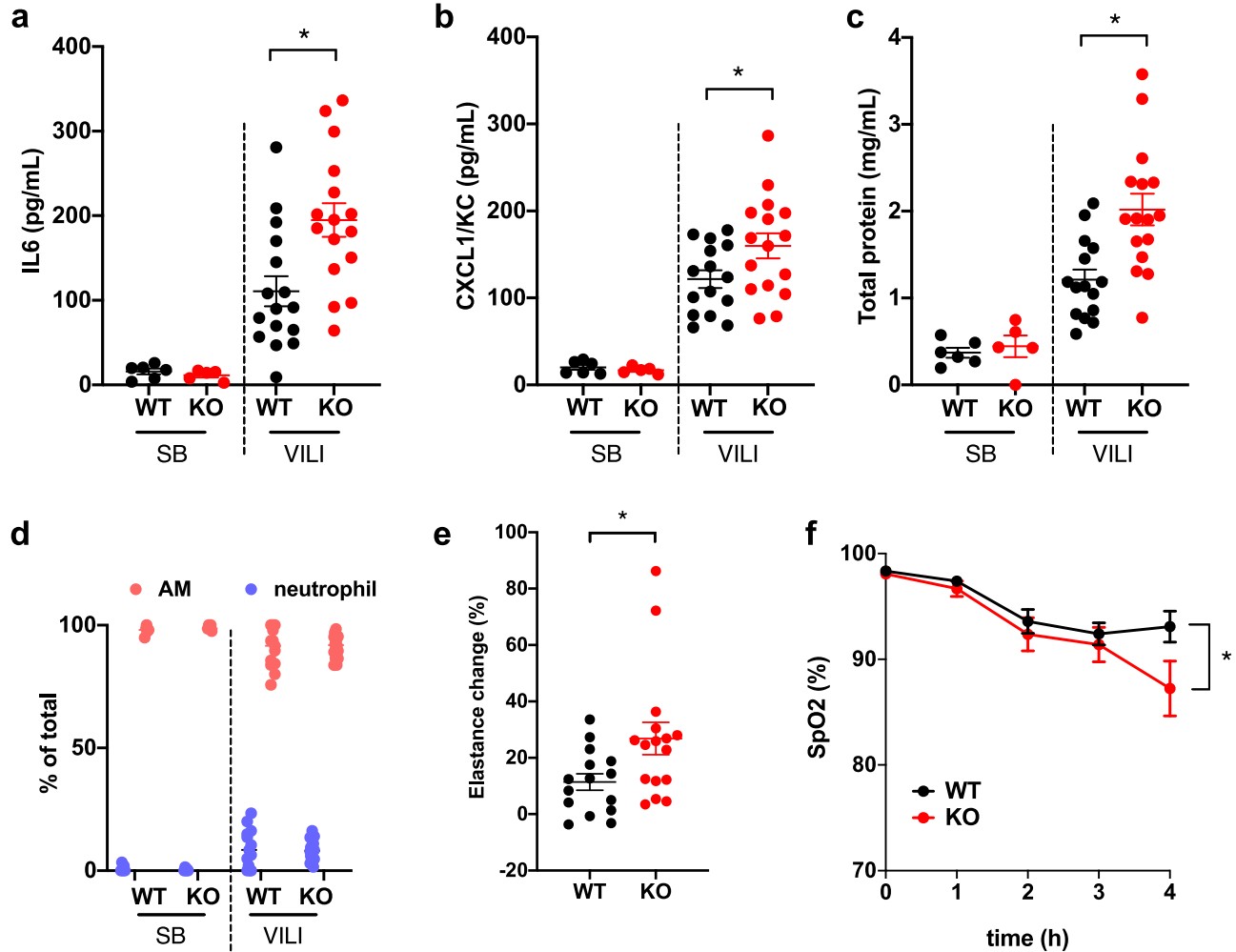

**Fig. 3 miR-146a knockout mice have increased injury during mechanical ventilation. a** BAL IL6 from spontaneously breathing (SB) and mechanically ventilated (VILI) wild-type (WT) or miR-146a knockout (KO) mice. Data normally distributed, $p = 0.0050$ comparing VILI WT to VILI KO via two-way ANOVA with post-hoc Tukey test. **b** BAL KC/CXCL1 from SB and VILI mice of either WT or KO. Data normally distributed, $p < 0.0001$ comparing VILI WT to VILI KO via two-way ANOVA with post-hoc Tukey test. **c** Bronchoalveolar lavage (BAL) protein concentrations from WT and KO mice subjected to VILI (or SB controls). Data normally distributed, $p < 0.0010$ comparing VILI WT to VILI KO via two-way ANOVA with post-hoc Tukey test. **d** BAL differential cell counts; $n = 16$ for KO VILI group, $n = 13$ for WT VILI group, $n = 5$ for WT SB, and $n = 5$ for KO SB groups. **e** Lung tissue elastance measurements during ventilation, normalized to initial values for each individual animal. Data non-normally distributed, $p = 0.0330$ via two-tailed Mann–Whitney test. **f** Oxygen saturation throughout duration of VILI measured via pulse oximetry. All data normally distributed, presented as mean + SEM, $p = 0.0091$ via repeated measures two-way ANOVA with Sidak's multiple comparisons test at 4 h time point; $n = 16$ for WT and KO VILI groups, $n = 6$ for WT SB, and $n = 5$ for KO SB groups for all panels except as noted above for **d**. Data are presented as mean ± SEM.

(mean ± SEM, $n = 2$, Supplementary Fig. 6c), and a near neutral zeta-potential ($-0.6 \pm -1.3$ mV, mean ± SD, $n = 5$). Encapsulation efficiency measured by agarose gel electrophoresis indicated nearly all of the pre-miR-146a was encapsulated in the LNP (Supplementary Fig. 6d). Treatment of primary AMs with these miR-loaded LNPs increased miR-146a levels ~100-fold (Fig. 5a). Since our previous work demonstrated that miR-146a regulates mechanotransduction in airway epithelial cells by targeting TRAF6 (ref. [24]), we also assessed TRAF6 expression in primary AMs following nanoparticle-based transfection. As shown in Fig. 5b, TRAF6 expression decreased following miR-146a transfection compared to scramble control. In addition, compared to cells transfected with a scramble-miR control, primary AMs transfected with miR-146a using lipofectamine (Fig. 5c) or LNP-based techniques (Fig. 5d) had dramatically decreased production of IL8 when subjected to in vitro barotrauma (oscillatory pressure). These data indicate that, similar to standard lipofectamine transfection, nanoparticle-based transfection of primary AMs can

be used to increase miR-146a to supraphysiologic levels. Although the 2–5-fold increase in endogenous miR-146a expression in AMs during in vitro barotrauma (Fig. 1b) is not sufficient to abrogate pressure-induced inflammation, increasing miR-146a levels 100-fold or more ameliorates pressure-induced IL8 release by AMs in vitro.

**miR-146a-loaded nanoparticles mitigate VILI in vivo**. To determine if increasing miR-146a levels in vivo holds therapeutic potential to reduce lung injury, we delivered pre-miR-146a-loaded LNPs or control nanoparticles loaded with a scramble pre-miR construct via intratracheal administration prior to injurious ventilation. Prior to these studies, the distribution of nanoparticle cellular uptake was examined using Cy3-labeled LNPs delivered by the intratracheal route followed by immunofluorescence imaging (Fig. 6a, b and Supplementary Fig. 7). Although only a subset of cells took up nanoparticles, co-localization analysis

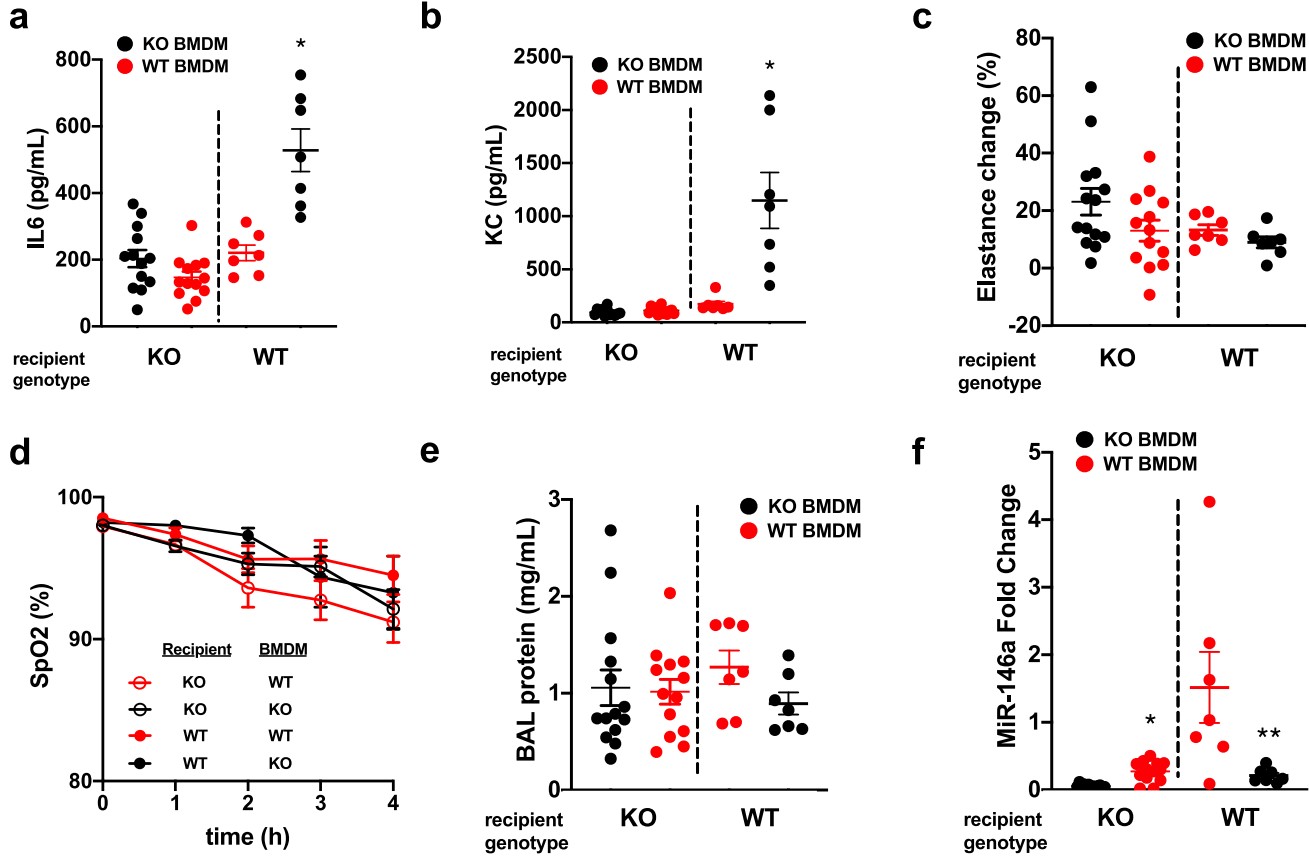

**Fig. 4 Adoptive transfer of macrophages alters miR-146a levels but is not sufficient to prevent lung injury. a** Bronchoalveolar lavage (BAL) IL6 levels from miR-146a knockout (KO, left panel) and wild-type (WT, right panel) mice subjected to ventilator-induced lung injury (VILI) following adoptive transfer of WT or miR-146a KO bone-marrow-derived macrophages (BMDMs). Data normally distributed, $p = 0.0847$ via two-tailed Student's $t$-test comparing KO recipients, $p = 0.0007$ via two-tailed Student's $t$-test comparing WT recipients. **b** BAL KC/CXCL1 levels from KO (left panel) and WT (right panel) mice subjected to VILI following adoptive transfer of WT or KO BMDMs. Data not normally distributed, $p = 0.0006$ via Mann–Whitney test. **c** Change in lung tissue elastance following 4 h VILI, normalized to baseline elastance prior to VILI. All data normally distributed, analyzed by two-tailed Student's $t$-test. **d** Oxygen saturation throughout duration of VILI measured via pulse oximetry. **e** BAL protein levels from KO (left panel) and WT (right panel) mice subjected to VILI following adoptive transfer of WT or KO BMDMs. **f** miR-146a expression in BAL cell pellet RNA from KO (left panel) and WT (right panel) mice subjected to VILI following adoptive transfer of WT or KO BMDMs. Relative expression determined by $\Delta\Delta$Ct method, normalized to WT mice that received WT BMDMs. Data normally distributed, analyzed by two-tailed Student's $t$-test; $*p < 0.0001$ compared to KO recipients that received KO BMDMs, $**p = 0.0290$ compared to WT recipients that received WT BMDMs. For all panels KO recipients: $n = 14$ KO BMDMs, $n = 13$ WT BMDMs. For WT recipients: $n = 7$ per group. Data are presented as mean ± SEM.

indicated that ~44% of the nanoparticles were delivered to epithelial cells while ~52% were delivered to AMs (Fig. 6c). However, the percentage of macrophage area that contained nanoparticles was significantly higher than the percentage of epithelial cell area that contained nanoparticles (Fig. 6d). AMs recovered from BAL fluid demonstrated an ~10,000-fold increase in miR-146a expression following delivery and ventilation while miR-146a levels in whole lung homogenate increased about 10–100-fold (Fig. 6e). Treatment with miR-146a nanoparticles dampened inflammation (Fig. 6f, g) and significantly reduced alveolar-capillary permeability (Fig. 6h). miR-146a-treated animals had significantly improved lung compliance compared to controls (Fig. 6i) and this was associated with improved oxygenation (Fig. 6j). Consistent with our in vitro data (Fig. 5b), TRAF6 mRNA levels were reduced in BAL cells following miR-146a delivery (Fig. 6k). We examined other known miR-146a targets and found decreased SMAD4 (ref. [30]) message levels in BAL cells, but no change in IRAK1 (ref. [24]) mRNA levels (Supplementary Fig. 8). No significant changes in miR-146a targets were noted in mRNA from whole lung homogenate (Supplementary Fig. 8). To assess whether the improvement in lung function was due to

altered recruitment of inflammatory cells, we measured differential cells counts and found no difference in the number of total BAL cells (Fig. 6l) or AMs (Fig. 6m). Interestingly, we did find fewer neutrophils in animals treated with miR-146a-loaded nanoparticles (Fig. 6n) indicating that miR-146a delivery reduces neutrophilic lung inflammation. These findings indicate that pulmonary delivery of miR-146a-loaded LNPs can be used to potently increase miR-146a levels and that this therapeutic increase mitigates lung injury during injurious MV.

## Discussion

Positive pressure MV is inherently injurious and molecularly targeted therapies that minimize VILI could improve clinical outcomes. In this study, we identified miR-146a as a mechanosensitive miR in AMs and elucidated the expression dynamics of miR-146a that might be required to mitigate lung injury during MV. Primary human AMs increase pro-inflammatory cytokine secretion and miR-146a expression when exposed to injurious mechanical forces in vitro (Fig. 1b). Following MV in vivo, murine BAL cells, which are >90% AMs in our model, also

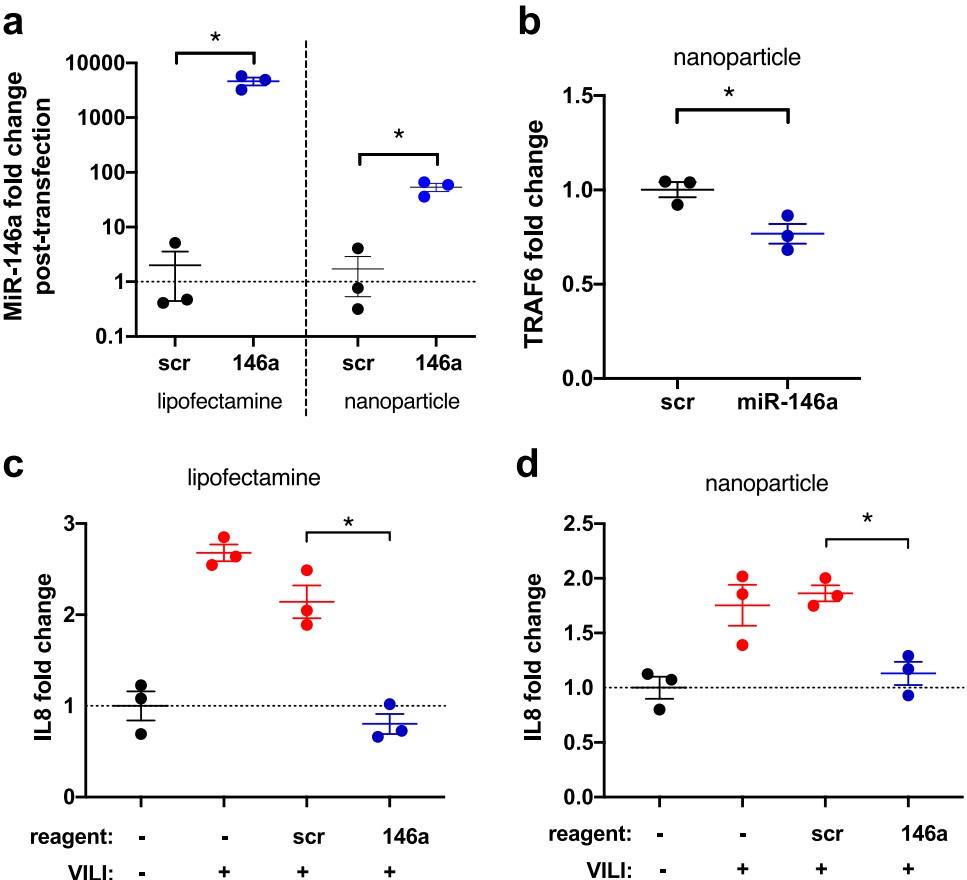

**Fig. 5 miR-146a overexpression dampens force-induced inflammation in vitro. a** miR-146a expression in alveolar macrophages (AMs) subjected to barotrauma following lipofectamine (left panel) or nanoparticle (right panel) transfection with pre-miR-146a, calculated by ΔΔCt method, normalized to scramble controls. Lipofectamine data log-normally distributed, $p = 0.0006$ via two-tailed Student's $t$-test comparing $\log_2$ (fold change), $n = 3$ per group. Nanoparticle data normally distributed, $p = 0.0048$ via two-tailed Student's $t$-test, $n = 3$ per group. **b** RelativeTRAF6 mRNA expression in AMs following nanoparticle mediated miR-146a transfection, calculated by ΔΔCt method, normalized to scramble transfected control. Data normally distributed, $p = 0.0240$ via two-tailed Student's $t$-test. $n = 3$ per group. **c** Fold change in IL8 secretion from AMs subjected to pressure (VILI) following scramble transfection or pre-miR-146a transfection using lipofectamine. Secretion was normalized to unpressurized control. Data normally distributed, $p = 0.0006$ comparing scr and 146a lipofectamine transfected groups, analyzed via one-way ANOVA with post-hoc Tukey test. $n = 3$ per group. **d** Fold change in IL8 secretion from AMs subjected to pressure (VILI) following scramble transfection or pre-miR-146a transfection using custom loaded lipid nanoparticles. Secretion normalized to unpressurized control. Data normally distributed, $p = 0.0135$ comparing scr and 146a nanoparticle transfected groups, analyzed via one-way ANOVA with post-hoc Tukey test; $n = 3$ per group. Data are presented as mean ± SEM.

exhibited increased miR-146a expression (Fig. 2b). Importantly, we demonstrated that MV in human subjects without ARDS also leads to increased miR-146a expression (Fig. 1e). Interestingly, mechanically ventilated patients with established ARDS had miR-146a levels that were similar to spontaneously breathing subjects raising the possibility that failure to upregulate miR-146a might exacerbate lung injury in the context of MV. Further experiments are needed to test this hypothesis.

The endogenous increases in miR-146a expression observed both in vitro and in vivo (2–10-fold, Figs. 1b, 2b) were not sufficient to inhibit the mechanotransduction processes responsible for proinflammatory cytokine production and lung injury during ventilation. In contrast, when standard lipofectamine or nanoparticle-based transfection techniques were used to significantly overexpress miR-146a in vitro by at least 100-fold, we observed significant inhibition of mechanically induced pro-inflammatory cytokine secretion in AMs (Fig. 5). Furthermore, our miR-loaded nanoparticles increased miR-146a expression in the lung in vivo (~1000–10,000-fold) and this increase dampened physiologic lung injury, pulmonary edema, and inflammation during injurious MV (Fig. 6). These data indicate that miR-146a upregulation in the setting of MV represents an

endogenous response that attempts to limit lung injury/inflammation. Although this endogenous response is insufficient to mitigate lung injury, increasing miR-146a to supraphysiologic levels is a novel strategy to decrease lung injury during MV.

The concept that a modest increase in a specific miR is insufficient to mitigate a pro-inflammatory insult has been reported previously. For example, cigarette smoke, which induces a neutrophilic inflammatory response, was shown to increase miR-135b by ~20-fold while a ~2000-fold increase in miR-135b expression was required to suppress cigarette-induced inflammation[31]. Similarly, we previously demonstrated that mechanically induced inflammation in primary human airway epithelial cells resulted in a modest twofold increase in miR-146a expression while an ~500–1000-fold increase was required to suppress mechanically induced inflammation in these epithelial cells[24]. Therefore, the modest increase of endogenous miR in response to different pulmonary insults may represent an insufficient-compensatory response, and appropriate delivery and dosing of miR may be critical for therapeutic applications.

Although miRs are attractive therapeutic candidates for a wide range of diseases[32], there is a unique set of challenges in delivering

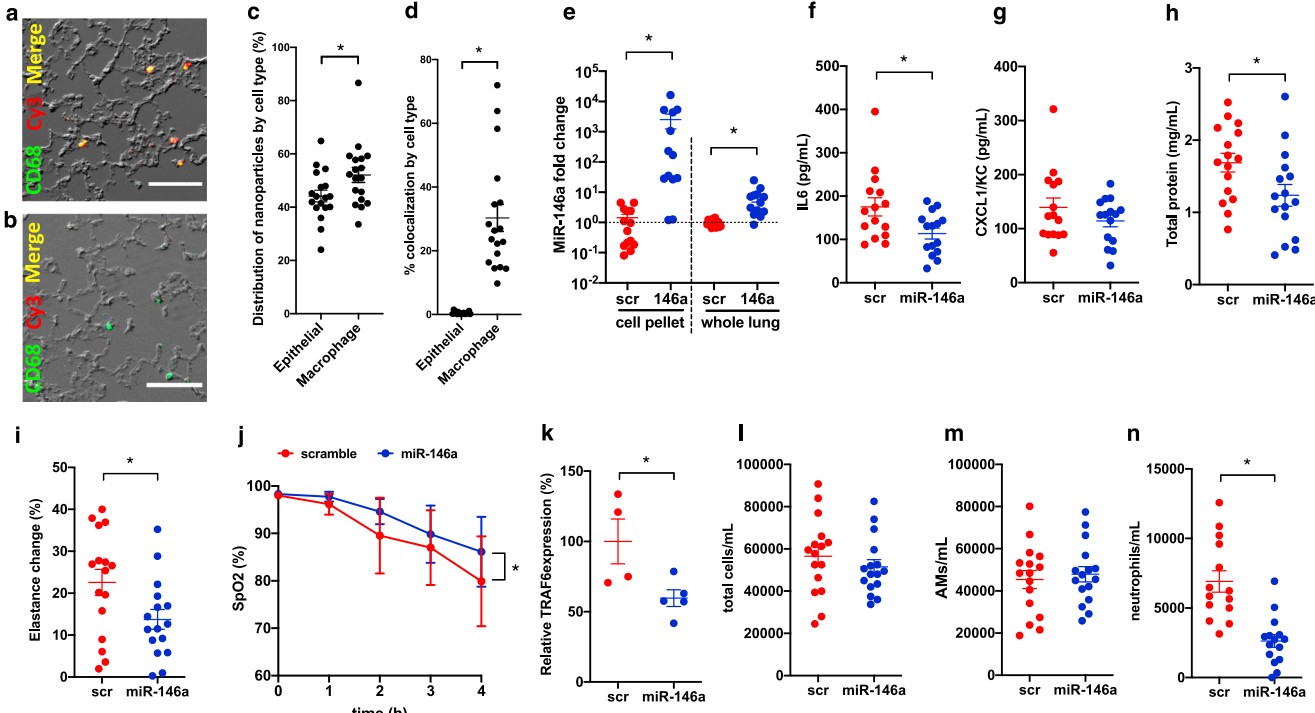

**Fig. 6 miR-146a-loaded nanoparticles mitigate lung injury during mechanical ventilation.** Representative fluorescence and DIC images from lung tissue of spontaneously breathing mice treated with Cy3-labeled nanoparticles (**a**) or PBS vehicle (**b**) prior to immunofluorescence staining for CD68 (green). Scale bar: 100 μm, n = 2 mice per group. **c** Distribution of Cy3-labeled nanoparticles by cell type using immunofluorescence images shown in Supplementary Fig. 7. Data normally distributed, p = 0.0323 via two-tailed Student's t-test; n = 18 high-power fields from two animals per group. **d** Percentage of epithelial and macrophage cell area containing Cy3-labeled nanoparticles. Data log normally distributed, p < 0.0001 via two-tailed Student's t-test on $\log_2$ transformed data. **e** miR-146a expression in RNA from BAL cell pellet (left panel) or lung tissue (right panel) following nanoparticle delivery and VILI. Relative expression determined by ΔΔCt method, normalized to VILI alone. Data log normally distributed, p < 0.0001 via two tailed Student's t-test on $\log_2$ (fold change). **f** BAL IL6 from miR-146a or scramble (scr) loaded nanoparticle treated mice following 4 h VILI. Data log normally distributed, p = 0.0132 via two-tailed Student's t-test on $\log_2$ transformed data. **g** BAL KC from miR-146a and scr mice following VILI. **h** BAL protein concentration BAL from miR-146a and scr mice following VILI. Data normally distributed, p = 0.0300 via two-tailed Student's t-test. **i** Change in lung elastance during 4 h period of ventilation. Data normally distributed, p = 0.0311 via two-tailed Student's t-test. **j** Oxygenation throughout duration of ventilation measured via pulse oximetry. Data normally distributed, p = 0.0110 via Sidak's multiple comparison test at 4 h time point following repeated measures two-way ANOVA. **k** TRAF6 message levels from BAL cells by ΔΔCt method normalized to scramble. Data log-normally distributed, p = 0.0356 via two-tailed Student's t-test on $\log_2$ (fold change); n = 4 for scramble group and n = 5 for miR group. **l** Total cell count obtained from BAL cell differential counts after ventilation. Data normally distributed, analyzed by two-tailed Student's t-test. **m** Alveolar macrophage (AM) cell count following ventilation. Data normally distributed, analyzed by two-tailed Student's t-test. **n** Neutrophil cell count following ventilation. Data normally distributed, p < 0.0001 via two-tailed Student's t-test. For all panels, n = 16 per group unless otherwise noted. Data are presented as mean ± SEM.

oligonucleotides to the distal lung[33]. First, although delivery to the upper airways via the oropharynx is straightforward, effective drug delivery to the distal airways and alveoli requires particle sizes on the nanometer scale[34]. Additional concerns include stability against degradation, inefficient cellular uptake, and off-target effects to other organs[33]. To overcome these challenges, several solutions have been proposed, including viral vectors and lipid-based nano-carriers. Viral vectors offer excellent stability and transfection efficiency but can induce undesirable inflammatory effects. LNPs are spontaneously assembled nanoparticles composed of amphiphilic lipids that can be loaded with DNA/oligonucleotides complexes[35]. The small size of LNPs is optimal for drug delivery to the alveolus, but they can suffer from poor transfection efficiency if their surface is highly negatively charged[36]. Recent formulations incorporate positively charged lipids and polymers to improve uptake efficiency and in this study we included cationic lipids in our nanoparticle formulation to neutralize the surface charge. AMs exhibit high phagocytic capacity and are thought to be the primary cells in the lung that take up and process nanoparticles[37]. As a result, previous studies have used antibody-based techniques to target miR delivery to other cells types and limit particle uptake by AMs[38]. However,

we identified AMs to be an important mechanosensitive cell type in VILI and therefore sought to deliver miR-146a to AMs directly. We therefore fabricated a non-antibody tagged LNP and demonstrated that these particles preferentially co-localize with a large fraction of AMs in the lung (Fig. 6a–d and Supplementary Fig. 7). LNPs loaded with pre-miR-146a were used to investigate if this delivery platform could modulate miR-146a expression in the mice (Fig. 6). To ensure a sufficient dose was delivered and time allowed for the pre-miR-146a construct to be processed to a mature miRNA, two doses of miR-146a encapsulated nanoparticles were given (24 h prior and immediately prior to MV). This technique increased miR-146a expression in the BAL cells (which are primarily AMs in our murine VILI model) by 1000–10,000-fold (Fig. 6e) and also increased miR-146a expression in the whole lung by 10–100-fold. This finding indicates that LNPs effectively increase miR-146a expression in AMs, and to a lesser degree increase miR-146a expression in other cell types.

Although we identified miR-146a expression in AMs as an important regulator of VILI, several other cell types also play an important role in lung injury during MV. Neutrophils contribute to the inflammatory response during MV[39] and we found that in vitro

barotrauma increased IL8 release in primary human neutrophils (Supplementary Fig. 2c). In contrast to the increase in miR-146a following barotrauma in AMs, we found that barotrauma in neutrophils did not consistently increase miR-146a levels (Supplementary Fig. 2d). In addition, the amount of miR-146a in neutrophils following barotrauma was nearly 100-fold lower than levels observed in AMs (Supplementary Fig. 2e) suggesting that AMs are a more important source of miR-146a than neutrophils in the setting of VILI. Pulmonary microvascular endothelial cells and alveolar/airway epithelial cells also respond to injurious forces during MV and may contribute to VILI[40]. Although the role of miR-146a in small airway epithelial cells has been investigated in vitro[24], future studies could investigate how modulating miR-146a expression in other cell types alters VILI. The current study focused on how miRs regulate mechanically induced inflammation during MV and how these mechanical forces can disrupt the alveolar-capillary barrier[40]. Previous studies indicate that modulating cytoskeletal structure and signaling can mitigate barrier disruption[41,42] and future studies could investigate how miRs, which are known to target these pathways, modulate VILI. In vivo, AMs interact directly with the alveolar microenvironment via attachment to alveolar epithelial cells[43]. Additional studies using in vitro co-culture models could be used to investigate if this cell–cell interaction regulates mechanically induced injury/inflammation. Lastly, although the use of primary human AMs in our study demonstrates the translational relevance of our in vitro and in vivo findings, these AMs may have different activation states, which may explain the variability in baseline cytokine expression between donors[43]. Larger scale studies using AMs isolated from a larger sample size of donor lungs could investigate the potential impact of polarization on AM contribution to VILI.

In summary, we have identified a novel pathway by which mechanotransduction in AM regulates VILI. In response to VILI, AMs modestly increase expression of miR-146a in an attempt to mitigate further lung injury. However, this force-induced increase in miR-146a expression is insufficient to dampen lung injury and higher levels of expression are necessary to mitigate force-induced injury. We have demonstrated that a novel nanoparticle-based delivery platform can be used to significantly increase miR-146a levels in vitro and in vivo and that this increase mitigates lung injury during MV.

## Methods

**Primary AM isolation and culture conditions**. Human AMs or PMA (phorbol 12-myristate 13-acetate)-differentiated THP-1 cells were used for in vitro experiments. THP-1 cells were differentiated for 48 h with 10 nM PMA. For experiments with primary cells, AMs were isolated via ex vivo lavage from de-identified lungs rejected for transplant and obtained from Lifeline of Ohio Organ Procurements agency (Columbus, OH). All lungs were from subjects with no history of chronic lung disease or cancer and were non-smokers for at least 1 year. After collection, RBCs were lysed and cells were enumerated and frozen down in FBS and 10% DMSO. Prior to experiments, AMs were rapidly thawed, added to RPMI (10% FBS, 1% antibiotic/antimycotic), and centrifuged at $400 \times g$ for 5 min. Total concentration and viability of cells were determined with trypan blue staining before seeding.

**Primary polymorphonuclear cell isolation and culture conditions**. Primary human PMNs were isolated from peripheral blood of healthy human donors by Ficoll density gradient separation method. Informed consent was obtained from all donors under an approved IRB protocol (IRB 2011H0059) in accordance with the Declaration of Helsinki. Briefly, 14 ml of Ficoll-Paque Premium sterile solution was added to the 35 ml blood/saline mixture slowly by a pipette, and then centrifuged at 1400 r.p.m. for 40 min at room temperature. The plasma was removed, and the PMN-rich red pellet was dissolved by adding cold 0.9% saline followed by cold 3% dextran, and incubated on ice for 30–40 min. After centrifugation, the upper PMN-rich layer was transferred to a new 50 ml conical tube, and 3× volume of cold 1× HBSS (without Ca & Mag & phenol red) was added. The PMN-rich cells were then pelleted by centrifugation at 1500 r.p.m. for 5 min at 4 °C and resuspended in 1× HBSS. Cell counts and viability were determined with trypan blue staining by automated cell counter. The purity of PMNs was assessed by differential cell count analysis following cytospin preparation. PMN cells were centrifuged again, and resuspended in phenol/FBS free RPMI supplemented with 1% penicillin/

streptomycin (P/S) and seeded on collagen-coated transwell permeable membrane inserts at a cell density of about $2 \times 10^6$ cells per insert. PMNs were allowed to attach for 30 min prior to the oscillatory pressure experiment. Cells without oscillatory pressure were used as a control. Cell viability after 16 h at air–liquid interface was assessed by live/dead staining using calcein and propidium iodide.

**Pressure apparatus**. Oscillatory pressure was applied to AMs at an air–liquid interface using a custom design apparatus[24]. Briefly, AMs were seeded on Transwell 6-well inserts at a density of about $1 \times 10^6$ cells per insert and allowed to adhere for 4 h in complete RPMI media supplemented with 10% FBS and 1% P/S at 37 °C and 5% $CO_2$. Media was exchanged and replaced with FBS-free RPMI supplemented with 1% P/S in the basal chamber and media was removed from the apical chamber. The well was sealed with rubber stoppers fixed to a central channel connected to the tubing. Tubes from each well were connected to a water manometer and a small animal ventilator (Harvard Apparatus, Holliston, MA, USA). Oscillatory pressure was applied for 16 h with a 0–20 cm-$H_2O$ range at 0.2 Hz. The media were subsequently collected for analysis by ELISA, and cellular RNA was isolated for RT-qPCR.

**Cell transfection**. AMs were seeded on 6-well Transwell inserts as described above for transfection[44]. Following adherence, AMs were transfected in Opti-MEM (with antibiotic supplement) with pre-miR-146a (Thermofisher Scientific, Waltham, MA, USA) loaded nanoparticles (200 nM miR) or lipofectamine (100 nM miR, Thermofisher Scientific) solution for 24 h. Following transfection, the media was replaced with RPMI and cells were subjected to pressure as described above.

**Animal use**. All animal studies were approved by the Institutional Animal Care and Use Committee (IACUC) at Ohio State under protocols 2011A00000081-R2 and 2013A00000105-R1. The studies were carried out in compliance with all relevant ethical regulations. MiR-146a KO mice[45] (stock: 016239) were purchased from the Jackson Laboratory (Bar Harbor, ME, USA) and crossed with in-house WT C57Bl/6J mice. KO and WT litter-mate controls were used for all experiments, with groups matched for age (age range 8–12 weeks) and sex.

**VILI protocol**. To induce VILI, mice were ventilated for 4 h with a tidal volume (TV) of 12 ml kg⁻¹ to induce volutrauma and 0 cm-$H_2O$ PEEP to induce atelectrauma using a Flexivent small animal ventilator (Scireq, Montreal, QC, Canada). Mice were anesthetized with ketamine and xylazine. Following induction of anesthesia, a tracheostomy cannula was placed, and mice were connected to the ventilator. Body temperature was maintained by using a heating pad beneath the animal. Baseline lung physiology measurements were obtained by performing a recruitment maneuver followed by forced oscillation to determine tissue elastance. Blood oxygenation was monitored using a mouse thigh pulse oximeter (Starr Lifesciences). At each hourly time-point, the lungs were recruited (two deep inflations) and lung physiology parameters were measured. Animals were volume resuscitated every 2 h ventilation with 10 μl g⁻¹ bolus saline. At the conclusion of ventilation or SB controls, mice were given a lethal overdose of anesthetic and BAL was performed by instilling 1 ml of PBS into the lungs twice and withdrawing it each time. BAL fluid was then centrifuged for 10 min at $500 \times g$ and the supernatant was collected for subsequent analysis. Red blood cells were lysed with RBC lysis buffer, and differential stain was performed (Hema 3 Stat Pack, Thermofisher Scientific) according to the manufacturer's instruction. The remaining cells were resuspended in TRIzol (Qiagen, Hilden, Germany) for RNA extraction. Following BAL, lungs were harvested and snap-frozen in liquid nitrogen. For whole lung gene expression, lungs were thawed, 750 μl of TRIzol was added, and homogenized using a handheld tissue homogenizer (Omni International, Kennesaw GA, USA).

**Clodronate depletion**. AMs were depleted using liposomal clodronate (Clodrosome, Brentwood, TN, USA). Mice were anesthetized with ketamine and xylazine and then administered 50 μl of 5 mg ml⁻¹ liposomal clodronate or unloaded liposomal controls via intratracheal instillation 48 h and 24 h prior to MV experiments.

**Bone-marrow-derived macrophage adoptive transfer**. BMDMs were isolated from the femur and tibia of 6–8-week-old WT or miR-146a KO mice. BMDMs were generated by differentiating bone marrow cells for 7 days in 20 ng ml⁻¹ MCSF[46]. Following differentiation, BMDMs were trypsinized, counted, and resuspended at $20 \times 10^7$ cells ml⁻¹. Fifty microliters of BMDM suspension ($1 \times 10^6$ total cells) was intratracheally instilled into mice 6–8 h prior to ventilation[47].

**Fabrication of miR-146a-loaded LNPs**. To overexpress miR-146a both in vitro and in vivo, we fabricated miR-146a-loaded LNPs. First, empty LNPs were fabricated using the ethanol dilution method where a mixture of lipids including 1,2-dioleoyl-sn-glycero-3-phosphoethanolamine (DOPE, Avanti Polar Lipids, Alabaster, AL, USA), 1,2-dioleoyl-3-trimethylammonium-propane (DOTAP, Avanti Polar Lipids), linoleic acid (LA, Sigma Aldrich, Natick, MA, USA), and D-α-Tocopherol polyethylene glycol 1000 succinate (TPGS, Sigma Aldrich) were diluted in ethanol in a molar ratio of 40:10:48:2 for DOPE:DOTAP:linoleic acid:TPGS,

respectively. This mixture was injected into 20 mM HEPES buffer and this solution was sonicated to form empty LNPs. Pre-miR-146a or pre-miR-scramble control (ThermoFisher Scientific) were mixed with PEI (2000 MW, Sigma Aldrich) at an N:P ratio of 10 (the ratio of moles of PEI amine groups to nucleic acid phosphate groups) in HEPES buffer and incubated at room temperature for 10 min to form PEI/miRNA polyplexes. The empty LNP and polyplex solutions were then mixed at a lipid to nucleic acid final mass ratio of 10 and this mixture was sonicated for 5 min at room temperature and then incubated for 10 min at room temperature. miR-146a-loaded LNPs were then concentrated to 1 nmol miR in 50 μL by spinning at $3500 \times g$ with 10K NMWL centrifugal filters (EMD Millipore, Burlington, MA, USA). Particle size, PDI, and zeta potential were measured via dynamic light scattering using NanoZS (Malvern). The morphology of the nanoparticles was assessed on a Thermo Scientific Glacios Cryo-TEM instrument. Briefly, Cryo-TEM samples were prepared by adding 3 μl of the miR-146a-loaded nanoparticles onto a 400 mesh copper lacey carbon grid. The grid was plunge-frozen in liquid ethane using the Vitrobot system. Images were acquired using Glacios CryoTEM. Encapsulation efficiency was determined by agarose gel electrophoresis[48]. Briefly, pre-miR-146a-loaded LNPs were lysed with 0.5% sodium dodecyl sulfate (SDS) (BioRad, Hercules, CA), and loaded on to 2% agarose (Fisher BioReagents) gel containing Labsafe nucleic acid stain (GBiosciences). Pre-miR-146a-loaded LNPs without SDS were also loaded to the gel for comparison. Gel electrophoresis was run at 100 V for 45–60 min in a 1× Tris-Acetate-EDTA (TAE) running buffer (Fisher BioReagents). Images of the gel were captured under UV light using ChemiDoc Imaging System (BioRad) and the amount of pre-miR-146a with and without SDS dissolution was quantitated. For in vitro experiments, cells were transfected with miR-loaded LNPs for 24 h prior to oscillatory pressure application. For in vivo experiments, 50 μl of miR-loaded LNPs (i.e. 1 nmol miR) were intra-tracheally instilled into mice 24 h and 0 h before ventilation.

*Immunofluorescence staining.* Fifty microliters of Cy3-labeled pre-miR negative control (Ambion, AM17120) loaded LNPs were administered intratracheally into WT mice. Control mice were treated with an equivalent volume of PBS without LNPs. Mice were euthanized 4 h after nanoparticle administration and the right heart was perfused with 1× $Ca^{2+}/Mg^{2+}$-free Dulbecco's phosphate-buffer saline (PBS) (Gibco) followed by 2% (v/v) paraformaldehyde (PFA) (Thermo Scientific) in PBS. Lungs were inflated to 20 cm $H_2O$ and then immersed in 2% PFA for 2 h. After fixation, lung tissues were embedded in optimal cutting temperature (OCT) compound (Fisher HealthCare) and frozen in 2-methylbutane (Honeywell) in a beaker submerged in liquid nitrogen. OCT embedded lungs were cryosectioned in the coronal plane at 7 μm using a cryostat microtome (Leica CM 1510S). Cryosections were fixed on slides with 2% PFA for 15 min at room temperature and washed several times with 1× PBS. Cryosections were permeabilized with 0.1% (v/v) Triton X-100 (Sigma) in PBS for 5 min, then incubated in blocking solution, 1% bovine serum albumin (BSA) (Fisher Scientific) in PBS, for 1 h at room temperature. The sections were immunostained for EpCAM (epithelial cells) and CD68 (macrophages) using fluorescently conjugated primary antibodies at a 1:50 dilution overnight at 4 °C. Rabbit anti-EpCAM antibody-Alexa Fluor 647 (ab237385), and rat anti-CD68 antibody-Alexa Fluor 488 (ab201844) were purchased from Abcam. Sections were washed three times with 1× PBS. Cryosections and coverslips were mounted with Immu-mount (Thermo Scientific). Images were acquired with a fluorescence microscope (Olympus IX81-ZDC), and processed using Fiji software.

*Nanoparticle cellular distribution analysis.* Nanoparticle distribution was quantitated by co-localization analysis of fluorescence images in MATLAB based on a previously published method[49]. Monochromatic images of each channel (epithelial cells-Alexa Fluor 647; macrophages-Alexa Fluor 488; nanoparticles-Cy3) were converted to 8-bit images. Pixel area of the epithelial cells and macrophages were computed by thresholding images and counting above threshold pixels. The threshold level was determined using control images without staining to remove background autofluorescence. Pixel area of red nanoparticles was measured in a similar manner. Nanoparticle distribution in epithelial cells was measured as pixel area overlapping between the purple and red channels while nanoparticle distribution in macrophages was measured as pixel area overlapping between green and red channels. The percentage of nanoparticle distribution was computed as the ratio of the epithelial or macrophage cell area with nanoparticle overlapped area to the total nanoparticle area. The percentage co-localization was calculated as the ratio of epithelial or macrophage cell area overlapping with red nanoparticle area to the total pixel area of epithelial cells or macrophages, respectively.

*Patient samples.* After informed consent was obtained, excess BAL fluid was obtained from patients with suspected infection during a clinically indicated bronchoscopy at The Ohio State Wexner Medical Center under an approved IRB protocol (IRB 2016H0009) in accordance with the Declaration of Helsinki. Potential subjects were identified by the pulmonary consult and intensive care unit teams. All patients ≥18 years of age were considered eligible. BAL was completed by the subject's treating physician and samples were sent to the clinical laboratory. Once all the ordered tests had been performed, excess BAL fluid was put on ice and transported to our laboratory. Total and differential cell counts were performed prior to centrifugation at $300 \times g$. The supernatant was aliquoted and frozen at −80 °C. TRIzol was added to the cell pellets for future RNA isolation.

**RNA extraction.** RNA was extracted for downstream analysis by RT-qPCR using the standard phenol–chloroform RNA extraction and purification was performed with TRIzol according to the manufacturer's protocol.

**ELISA.** Enzyme-linked immunosorbent assays (ELISA) were performed to assess secreted cytokine/mediator levels in media or BAL. For human cytokines, OptEIA kits (BD Biosciences, Franklin Lakes, NJ, USA) were used (IL6, IL8, IL1β, TGFβ, IL12) and the manufacturer's protocol was followed. For mouse mediators, Duoset ELISA kits (R&D Systems, Minneapolis, MN, USA) were used (IL6, CXCL1/KC) and the manufacturer's protocol was followed.

**RT-qPCR.** RT-qPCR with TaqMan primer/probes was used to determine relative expression levels of miR-146a compared to U18 or U6 (human) or sno251 (mouse) endogenous control genes (ThermoFisher, assay IDs: 478399_mir, 001204, 001093, 001236, respectively). TRAF6 relative expression was determined relative to GAPDH control (ThermoFisher, assay IDs: Hs00939742_g1 (TRAF6) Hs02786624_g1 (GAPDH) for human samples, and Mm00493836_m1 (TRAF6), Mm99999915_g1 (GADPH) for mouse). Following RNA extraction, cDNA was synthesized using a high-capacity cDNA transcription kit (Thermofisher). qPCR was then performed with Taqman Master Mix (Thermofisher) on Roche Light-Cycler480 and data were quantified by the ΔΔCT method.

**Statistics.** For laboratory-based experiments, statistical analysis was performed using Graphpad Prism 8. The distribution of all data was tested for normality via a Shapiro-Wilk test. All data are presented as mean + SEM except where noted otherwise. Experimental sample size was determined by performing a pilot experiment with six animals and the sample mean and standard deviation estimates were used to calculate the final sample size with a power of 0.8 and $\alpha = 0.05$. To compare two groups with normal distribution, a Student's $t$-test was used for comparisons or $t$-test on log-transformed data if data were log-normally distributed. A Mann–Whitney test was used to the compare ranks if the data were not normally distributed. For analysis of qPCR data, the statistical comparison was performed on log-transformed fold change data. For comparison among multiple groups, all data were normally or log-normally distributed and an ANOVA was performed on log-transformed or untransformed data as appropriate. For experiments with two-independent variables, a two-way ANOVA was performed after testing data for normality as described. If a significant effect was determined by ANOVA, a Tukey post-hoc test for individual group comparisons was performed. A $p$-value of <0.05 was considered statistically significant. Outliers were identified using non-linear regression via the ROUT method with a $Q$ threshold of 1%[50]. For the clinical data, association analyses between pairs of variables were conducted with Fisher's exact tests (for categorical variables) and two-tailed $t$-tests or Kruskal–Wallis tests (for continuous variables as appropriate based on the normality of the data) using SAS version 9.4 (Cary, NC). Generalized linear models were used to assess for an association between variables with more than two groups for comparison.

**Reporting summary.** Further information on research design is available in the Nature Research Reporting Summary linked to this article.

## Data availability
The data that support the findings of this study are available from the corresponding authors upon reasonable request. Source data are provided with this paper.

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

## Acknowledgements

This work was partially supported by NIH Grants K08 GM102695 (J.A.E.), R56 HL142767 (J.A.E., S.N.G.), R01 HL076278 (M.D.W.), a Department of Defense Grant W81XWH-19-1-0210 (S.N.G., J.A.E.), and an OSU Presidential Fellowship (C.B.). We would like to thank the analytical flow cytometry core for their technical assistance, the Davis Heart and Lung Research Institute, and the Center for Electron Microscopy and Analysis at The Ohio State University.

## Author contributions

C.B., S.N.G., and J.A.E. conceived and designed the study. C.B. and Q.F. performed the majority of experiments and analyzed the data under the supervision of S.N.G. and J.A.E. Additional data were acquired and analyzed by V.S., H.L., P.P., C.S., M.T., M.D.W., and R.K.P. Data were interpreted by C.B., Q.F., M.D.W., J.W.C., M.N.B., R.J.L., S.N.G., and J.A.E. The manuscript was drafted by C.B., M.N.B., S.N.G., and J.A.E. Critical revision of the manuscript was performed by all authors.

## Competing interests

The authors declare no competing interests.
