## [Peer Review File · Nature Communications]

Reviewers' comments:

Reviewer #1 (Remarks to the Author):

The manuscript by Bobba et al. investigates a mechanosensitive microRNA, miR-146a, and its expression by alveolar macrophages that acts as a protective mechanism from lung injury caused by mechanical ventilation. The investigators also evaluate a nanoparticle delivery method that induces enhanced miR-146a overexpression to further validate its protective ability in mechanical lung injury. The authors utilized in-vitro systems, mouse models, and biospecimens from mechanically ventilated patients to examine the possible protective role of miR-146a from mechanical lung injury. They measured miR-146a expression in human AMs and BAL cells in response to oscillatory transmural pressure and in mechanically ventilated patients, with and without ARDS; although very little was discussed or performed in the context of ARDS following the initial figure. Nevertheless, the authors thoroughly and successfully demonstrate the role of the mechanosensitive microRNA, miR-146a, as a protective mechanism to mitigate ventilator-induced lung injury via knockout and overexpressing studies. Their results indicate that alveolar macrophages have little endogenous contribution, as shown in the liposomal clodronate studies and the adoptive BMDM transfer studies. Nevertheless, miR-146a overexpression in alveolar macrophages in vitro and in vivo mitigated some of the negative outcomes in these models. While overexpression of miR-146a in the lung appears to confer some protection against mechanical injury, the mechanism for this protection still lacks great clarity and the contribution from cell types other than alveolar macrophages appears probable. However, the studies show the use of miR-146a in translational studies may be a promising avenue to pursue once the mechanism is further investigated in future study.

Major Comments:

1. The comparison of miR-146a expression from primary human AMs subjected to an In vitro barotrauma model and BAL cells isolated from mechanically ventilated need additional clarification. Are the same forces and injury outcomes produced in the in vitro barotrauma model similar to observations in the clinic of mechanically ventilated patients, with and without ARDS?
2. The authors implement a previously described in vitro model of barotrauma using primary human AMs; however, they also characterize miR-146a expression from BAL cells collected from spontaneous breathing and mechanically ventilated patients. This mixture often includes alveolar macrophages, neutrophils, and other myeloid immune cells. If this mixture of cells were isolated and subjected to the same in vitro model of barotrauma, would they produce similar changes in miR-146a expression and inflammatory responses as the primary human AMs? This should be added to the discussion.

3. While it's possible that the failure to see statistically significant differences in miR-146a expression between vehicle-control and liposomal clodronate groups could be due to variability between clodronate and vehicle treated animals, it appears that it could also be due to the neutrophil influx that also accumulates in the BAL cell collection in the VILI groups. This should be investigated or explained.

4. The authors incorrectly identify cells as Alveolar macrophages in the adoptive BMDM transfer studies, while they should be labeled as BMDMs. There was no evidence or data to support that these adoptively transferred BMDMs localized into the airways and differentiated into AM populations. This limitation should be addressed in the manuscript.

Minor Comments:

1. Authors claim "Although miR-146a expression was increased in mechanically ventilated patients without ARDS, levels in mechanically ventilated patients with ARDS were similar to spontaneous breathing controls raising the possibility that failure to upregulate miR-146a could exacerbate lung injury in the context of mechanical ventilation." There does not seem to be enough evidence provided to make this claim.

2. As mentioned above, it would be interesting to see how miR-146a expression changes in a "two-hit" VILI model that is more similar to clinical ARDS and to determine which cell types fail to express miR-146a in that model compared to the murine VILI model to further validate the above claim. It would also be interesting to see if experimentally overexpressing miR-146a in both in vitro and in vivo "two-hit" VILI models also attenuates the negative outcomes associated with those models.

3. The predominate cell type in the murine BAL cell collections were determined to be macrophages. Why was this not performed on the human BAL cell collections to further validate the connection between primary human AMs and human BAL cells from mechanically ventilated patients?

4. Please explain what the the difference in miR-146a expression between human BAL cells and murine BAL cells could suggest in terms of endogenous protective mechanisms, especially in the context of lung injury modelling. What implications or limitations could this difference suggest for other murine ALI models?

5. The authors claim that the lack of statistical difference between vehicle-control and liposomal clodronate groups could partially be due to variability between clodronate and vehicle treated animals. Could this be further explained?

6. In the results section that discusses Figure 6, the authors mistakenly address lung elastance for lung compliance. These mechanical properties are not synonymous. The lung elastance is the reciprocal of lung compliance; therefor the graphs and descriptions do not agree.

7. Is there supporting evidence that indicates the VILI model (tidal volume = 12 CC/kg, PEEP = 0, duration = 4 hours) produces substantial indications of ALI? While the increase in IL-6 seems adequate, the change in BAL protein seems minor. Is there supporting histology or histological assessments to further validate substantial induction of ALI?

8. Mice were anesthetized with ketamine and xylazine; however, it has been recommended to avoid using these drugs when looking at VILI and inflammatory models as they can alter inflammation (Yang J, Li W, Duan M, Zhou Z, Lin N, Wang Z, Sun J, Xu J Inflamm Res. 2005 Mar; 54(3):133-7.). Please explain why these were chosen. Could this partially explain why there was no difference seen in the number of recruited immune cells in the in vivo studies?

Reviewer #2 (Remarks to the Author):

Acute respiratory distress syndrome (ARDS) is a major clinical problem induced by trauma and infections of the lung and treatment approaches are often limited and ineffective. The authors have observed that by increasing levels of miR-146a in macrophages (potentially) by employing a nanoparticle-based delivery platform (transfection systems) was sufficient to prevent tissue injury and alterations in lung function in models of mechanical induced lung injury. Notably, the authors have previously shown increased miR-146a expression in primary human lung epithelial cells exposed to cyclic transmural pressure. Furthermore, overexpression of miR-146a was shown to reduced pressure-induced inflammation in human epithelial cells (EC) by modulating the signaling molecules linked to toll-like receptors. Like EC, macrophages have also been linked to the pathophysiology of mechanical induced aspects of ARDS. The authors extended their studies and investigate the role of miR-146a in mechano-transduction processes induced in AMs. This an excellent paper, well-constructed and controlled and the stated conclusions are largely supported by the data.

Supp'- Figure 1. Specificity to IL-8 (increased levels) is surprising given one may expect a more global change in inflammatory status in response to MV induced trauma. Please comment

Figure 1. "To determine if force induced miR-146a is observed

clinically, miR-146a was measured in BAL cells obtained from a cohort of patients that underwent a clinically indicated bronchoscopy" What is the cellular make-up of the BALF- e.g. neutrophil to macrophage numbers?

Figure 2. In my opinion a 10-fold increase in the levels of a miR is more than a modest increase in the airways under such conditions. This is a very important correlative figure implicating MiR-146a to physiological function.

Figure 3. This data shows an important protective role for miR146a in the injury response to ventilation. However, what happens after the acute phase (4 hours)? Especially to SPO₂ and elastance? Can the authors provide data or comment on the development of lesion and pathophysiology in the more chronic phase?

Fig 4 and 5. The data would be strengthened if you could show a change in activation status (proinflammatory response) of the KO macrophages before and/or after transfer And in the specific uptake of miR-146a by macrophages after transfection.

Fig. 5. lipofectamine or nanoparticle (i.e. cationic lipoplex) based transfection techniques were used to overexpress miR-146a by 100-fold in primary human AMs (Figure 5A). – this is increasing the amount of miR-146a in these cells NOT increasing the expression.

Fig. 5. And 6. What is the percentage of transfection of macrophages in the lung of mice? Are they the major cell where miR146 levels are increased?

Reviewer #3 (Remarks to the Author):

This is an interesting paper with a focus on the biology of miR-146a and its role in lung macrophages mechanotransduction and lung injury. Overexpression of miR-146a has clear effects in the studies presented, but it is not clear how relevant this potential treatment would be for clinical use. There is also a tight focus on alveolar macrophages (AMs), but many other cell types may be affected by miR-146a overexpression in vivo, not to mention the various pathways that miRNAs in general can hit which further complicate the studies. It would also be nice to see an animal model with a clear readout for efficacy; something to show that this therapy makes a meaningful impact on animal health or survival, as well as to put the therapy in context with a clear clinical unmet need. It seems that this paper is highly relevant for a specialized readership, but not of sufficiently broad interest for a multidisciplinary journal due to the tight focus on miR-146a and the modest demonstration of therapeutic utility with a nanoparticle that is not likely to be useful clinically.

1. How can one be sure the effects (Figure 6) are due to AMs and not other cells?

2. In addition to TRAF6, miR-146a likely silencing additional genes. One should look at gene signatures following miR-146a delivery more broadly to understand globally what all targets are inhibited. Some of these may be having a larger effect than TRAF6 or there could be combinatorial effects. There are many papers on miRNAs showing broad effects on gene expression broken into various pathways and cellular programs (bioinformatic analysis, clustering, etc.). This is likely relevant to the effects seen and lung injury treatment and should be studied.

3. There is very little discussion of the delivery vehicle details in the main text (chemistry used, specific characteristics, biophysical properties, etc.). The experimental section details a procedure whereby miRNA was complexed with PEI2000, which would initially make uncontrolled polyplexes of various diameters. Separately, "empty liposomes" were made by combining DOPE, DOTAP, linoleic acid, and TPGS in HEPES buffer. Given that these molecules are insoluble in water (HEPES buffer), it is confusing how this process could be conducted. Were these water insoluble lipids sonicated to form liposomes or lipid nanoparticles (LNPs)? Were they actually prepared by dissolving in an organic, water miscible solvent like ethanol, to then combine with HEPES buffer (the "ethanol dilution method")? Nevertheless, it is unclear what was actually made. Imaging by electron microscopy (e.g. TEM) could help tell if one made liposomes (with a bilayer) or LNPs (with a solid core). This is critical because the next step involves "loading" PEI/miRNA lipoplexes inside of presumably empty liposomes simply by incubating. It is also unclear how this would happen. One could imagine electrostatic interactions between anionic miRNAs and cationic materials (PEI, DOTAP, etc.), but what would be a driving force to load polyplexes inside of liposomes? Could liposomes / lipoplex complexes have been made instead? More details and characterization are needed.

4. There is a discussion about how size can affect pulmonary delivery, and a sentence that says the particles used had an average diameter of 123nm. Is this a suitable size? What about PDI / uniformity, miRNA encapsulation, dose? 77% encapsulation efficiency is reported in the SI, but how much was attempted to be loaded? There is a lack of details in the main text about the delivery experiment, readouts, etc.

5. It is unclear how much miRNAs were packaged and delivered in vivo. It is also unclear how much miRNA reached targeted cells following intratracheal installation. Some characterization of dosage is needed. And some evidence that the miRNAs are actually in the AMs is needed as well. One could do this by fluorescence, radiolabeling, or other methods to track miRNA and/or nanoparticle uptake in lung cells in vivo.

We would like to thank all three reviewers for their insightful comments and detailed review of our manuscript. We have addressed all of the concerns/comments raised by the reviewers in a point-by-point fashion below. We performed additional experiments and generated multiple new figures of data in response to specific critiques from the reviewers including data on the response of neutrophils to mechanical stress, the structure of the lipid nanoparticle used for in-vivo delivery, and distribution of this nanoparticle in the lungs. These data as well as additional clarifications and justifications have strengthened the manuscript and will lead to a more impactful publication.

Reviewer #1 (Remarks to the Author):

The manuscript by Bobba et al. investigates a mechanosensitive microRNA, miR-146a, and its expression by alveolar macrophages that acts as a protective mechanism from lung injury caused by mechanical ventilation. The investigators also evaluate a nanoparticle delivery method that induces enhanced miR-146a overexpression to further validate its protective ability in mechanical lung injury. The authors utilized in-vitro systems, mouse models, and biospecimens from mechanically ventilated patients to examine the possible protective role of miR-146a from mechanical lung injury. They measured miR-146a expression in human AMs and BAL cells in response to oscillatory transmural pressure and in mechanically ventilated patients, with and without ARDS; although very little was discussed or performed in the context of ARDS following the initial figure. Nevertheless, the authors thoroughly and successfully demonstrate the role of the mechanosensitive microRNA, miR-146a, as a protective mechanism to mitigate ventilator-induced lung injury via knockout and overexpressing studies. Their results indicate that alveolar macrophages have little endogenous contribution, as shown in the liposomal clodronate studies and the adoptive BMDM transfer studies. Nevertheless, miR-146a overexpression in alveolar macrophages in vitro and in vivo mitigated some of the negative outcomes in these models. While overexpression of miR-146a in the lung appears to confer some protection against mechanical injury, the mechanism for this protection still lacks great clarity and the contribution from cell types other than alveolar macrophages appears probable. However, the studies show the use of miR-146a in translational studies may be a promising avenue to pursue once the mechanism is further investigated in future study.

Major Comments:

1. The comparison of miR-146a expression from primary human AMs subjected to an In vitro barotrauma model and BAL cells isolated from mechanically ventilated need additional clarification. Are the same forces and injury outcomes produced in the in vitro barotrauma model similar to observations in the clinic of mechanically ventilated patients, with and without ARDS?

There are three primary biomechanical mechanisms by which mechanical ventilation causes lung injury. Injury can occur from excessive stretch of lung cells (*i.e.* volutrauma), stress from collapse and reopening of fluid filled lung units (*i.e.* atelectrauma), and compressive stress from increased transmural pressure (*i.e.* barotrauma). It is known that barotrauma and the other injurious forces occur in patients undergoing mechanical ventilation, but because these injuries are occurring throughout the lungs on a cellular scale, clinicians do not have the ability to determine the degree to which each of these injuries are present in each patient undergoing

mechanical ventilation. Clinicians use strategies such as ventilating patients with low tidal volumes (*i.e.* 6-8 cc/kg predicted body weight), keeping inspiratory airway pressures less than 30 cm H₂O, and the use of positive end expiratory pressure (PEEP) to try and minimize the injury caused by all three mechanisms. However, prior work has demonstrated that injury cannot be avoided completely.(1, 2) Inflammatory cytokines such as IL8 are also increased in the bronchoalveolar lavage fluid of mechanically ventilated patients and this is recapitulated in our in vitro models. Therefore, it is highly likely that the injurious forces occurring in the in vitro barotrauma model are also present in our mechanically ventilated patients both with and without ARDS.

2. The authors implement a previously described in vitro model of barotrauma using primary human AMs; however, they also characterize miR-146a expression from BAL cells collected from spontaneous breathing and mechanically ventilated patients. This mixture often includes alveolar macrophages, neutrophils, and other myeloid immune cells. If this mixture of cells were isolated and subjected to the same in vitro model of barotrauma, would they produce similar changes in miR-146a expression and inflammatory responses as the primary human AMs? This should be added to the discussion.

We thank the reviewer for this insightful question and suggestion. Unfortunately, the BAL biospecimens used in our study were collected several years ago and frozen upon collection precluding our ability to subject freshly isolated cells to our in vitro VILI model. Although the macrophage response to mechanical stress was described in our initial submission of this manuscript, we agree that evaluating the neutrophil response to mechanical stress is warranted given that the BAL cells from our mechanically ventilated subjects contain a mixed inflammatory population comprised of macrophages and neutrophils (Table 1). Since we were unable to examine the contribution of neutrophils to pro-inflammatory cytokine and miR-146a expression using BAL cells from mechanically ventilated subjects, we isolated primary human neutrophils from healthy donors (Supplemental Figures 2A-B) and subjected them to our in vitro barotrauma model. We found that in vitro barotrauma increased IL8 release in primary human neutrophils to varying degrees depending on the donor (Supplemental Figure 2C). In contrast to the upregulation of miR-146a following barotrauma in macrophages, we found that barotrauma in neutrophils did not consistently increase miR-146a levels (Supplemental Figure 2D). Furthermore, the amount of miR-146a in neutrophils following barotrauma was nearly 100-fold lower than in alveolar macrophages (Supplemental Figure 2E). Therefore, we conclude that the miR-146a responses from BAL cells is primarily a macrophage response. These data have been added to the Results and Discussion sections and are shown in the new Supplemental Figure 2.

3. While it's possible that the failure to see statistically significant differences in miR-146a expression between vehicle-control and liposomal clodronate groups could be due to variability between clodronate and vehicle treated animals, it appears that it could also be due to the neutrophil influx that also accumulates in the BAL cell collection in the VILI groups. This should be investigated or explained.

We thank the reviewer for this insightful comment. Our newly added data (see response to comment #2 above) demonstrate that in vitro barotrauma in primary human PMNs does not consistently increase miR-146a levels and when it does the amount of this miR is nearly 100-fold lower than in alveolar macrophages. These data suggest that it is unlikely that neutrophil influx

is responsible for the differences in miR-146a levels in clodronate treated mice subjected to mechanical ventilation.

4. The authors incorrectly identify cells as Alveolar macrophages in the adoptive BMDM transfer studies, while they should be labeled as BMDMs. There was no evidence or data to support that these adoptively transferred BMDMs localized into the airways and differentiated into AM populations. This limitation should be addressed in the manuscript.

This is an excellent point and we have edited the manuscript as suggested.

Minor Comments:

1. Authors claim “Although miR-146a expression was increased in mechanically ventilated patients without ARDS, levels in mechanically ventilated patients with ARDS were similar to spontaneous breathing controls raising the possibility that failure to upregulate miR-146a could exacerbate lung injury in the context of mechanical ventilation.” There does not seem to be enough evidence provided to make this claim.

We agree with the reviewer and have removed the statement from the Results section. We now raise this possibility in the Discussion and clearly state that further experiments are needed to test this hypothesis.

2. As mentioned above, it would be interesting to see how miR-146a expression changes in a “two-hit” VILI model that is more similar to clinical ARDS and to determine which cell types fail to express miR-146a in that model compared to the murine VILI model to further validate the above claim. It would also be interesting to see if experimentally overexpressing miR-146a in both in vitro and in vivo “two-hit” VILI models also attenuates the negative outcomes associated with those models.

We agree with the reviewer that it would be interesting to examine the role of miR-146a in a two-hit models that are more similar to clinical ARDS conditions and are planning to examine this in the future.

3. The predominate cell type in the murine BAL cell collections were determined to be macrophages. Why was this not performed on the human BAL cell collections to further validate the connection between primary human AMs and human BAL cells from mechanically ventilated patients?

Differential cell count data from human BAL specimens can be found in Table 1. See response to Major Comments #2 above.

4. Please explain what the the difference in miR-146a expression between human BAL cells and murine BAL cells could suggest in terms of endogenous protective mechanisms, especially in the context of lung injury modelling. What implications or limitations could this difference suggest for other murine ALL models?

Patients undergoing mechanical ventilation have a ~4-fold increase in miR-146a levels on average compared to spontaneously breathing patients (Figure 1E). Wild-type mice subjected to

VILI have ~10-fold increase on average in miR-146a levels compared to spontaneously breathing control mice (Figure 2B). Changes in miR-146a levels are measured by qPCR and are normalized to the control group. As a result, absolute differences cannot be compared between the 2 different experiments and we are therefore unable to draw any meaningful conclusions about the differences between human and mouse miR-146a levels.

5. The authors claim that the lack of statistical difference between vehicle-control and liposomal clodronate groups could partially be due to variability between clodronate and vehicle treated animals. Could this be further explained?

Because the studies in Supplemental Figure 4 contain 2 independent variables (i.e. VILI vs SB controls & clodronate vs vehicle) the data were analyzed by 2-way ANOVA. If you analyze the difference between miR-146a levels (Supplemental Figure 4H) only in the VILI groups using a non-parametric t-test (Mann-Whitney) the difference between vehicle and clodronate groups is significant with a p-value of 0.0256. However, when the data are analyzed by 2-way ANOVA and adjusted for multiple comparisons the difference is not statistically significant with an adjusted p-value of 0.119. The fact that the 2-way ANOVA (which includes all 4 groups) is not significant but the t-test is significant suggests that the variability in miR-146a levels between groups limits the ability to detect statistically significant differences.

6. In the results section that discusses Figure 6, the authors mistakenly address lung elastance for lung compliance. These mechanical properties are not synonymous. The lung elastance is the reciprocal of lung compliance; therefor the graphs and descriptions do not agree.

We have changed the wording as suggested by the reviewer.

7. Is there supporting evidence that indicates the VILI model (tidal volume = 12 CC/kg, PEEP = 0, duration = 4 hours) produces substantial indications of ALI? While the increase in IL-6 seems adequate, the change in BAL protein seems minor. Is there supporting histology or histological assessments to further validate substantial induction of ALI?

The pathophysiologic hallmarks of lung injury during ARDS are compromise of the alveolar capillary barrier and inflammation that lead to impaired lung function. Impaired lung function manifests as increased lung stiffness (i.e. elastance) and impaired oxygenation. As shown in Figure 2, the mice in our studies develop each of these features of lung injury including increased inflammatory mediators (i.e. BAL IL6 and KC) and increased barrier permeability (i.e. BAL protein levels). This finding is highly statistically significant with a p value of <0.0001. Notably, BAL protein levels increase by greater than 100% following ventilation. Although we did not perform a detailed histologic assessment of lung injury for this study, we and other groups have shown that similar ventilator settings lead to histologic lung injury.(3-5)

8. Mice were anesthetized with ketamine and xylazine; however, it has been recommended to avoid using these drugs when looking at VILI and inflammatory models as they can alter inflammation (Yang J, Li W, Duan M, Zhou Z, Lin N, Wang Z, Sun J, Xu J Inflamm Res. 2005 Mar; 54(3):133-7.). Please explain why these were chosen. Could this partially explain why there was no difference seen in the number of recruited immune cells in the in vivo studies?

The anesthetics chosen for our experiments were based on recommendations from the university veterinarians at the time our IACUC protocol was developed. The paper referenced by the reviewer showed that high dose ketamine (i.e 50 mg/kg) administered to rats that received IV LPS had decreased IL6 and TNF alpha levels at 2 hours. We used a ketamine dose that was 5-fold lower (i.e 10 mg/kg) than the dose used in the manuscript above. Interestingly, the dose of ketamine in the above manuscript that was closest to the dose we used (5mg/kg) did not alter IL6 or TNF levels. Furthermore, other investigators have used ketamine as an anesthetic in models of ventilator induced lung injury and have observed differences in lung inflammation between experimental groups.(6-8) Given that our mice undergoing mechanical ventilation and the spontaneously breathing control mice both received ketamine, it is unlikely that this accounts for the lack of differences in recruited immune cells.

Reviewer #2 (Remarks to the Author):

Acute respiratory distress syndrome (ARDS) is a major clinical problem induced by trauma and infections of the lung and treatment approaches are often limited and ineffective. The authors have observed that by increasing levels of miR-146a in macrophages (potentially) by employing a nanoparticle-based delivery platform (transfection systems) was sufficient to prevent tissue injury and alterations in lung function in models of mechanical induced lung injury. Notably, the authors have previously shown increased miR-146a expression in primary human lung epithelial cells exposed to cyclic transmural pressure. Furthermore, overexpression of miR-146a was shown to reduced pressure-induced inflammation in human epithelial cells (EC) by modulating the signaling molecules linked to toll-like receptors. Like EC, macrophages have also been linked to the pathophysiology of mechanical induced aspects of ARDS. The authors extended their studies and investigate the role of miR-146a in mechano-transduction processes induced in AMs. This an excellent paper, well-constructed and controlled and the stated conclusions are largely supported by the data.

We thank the reviewer for this feedback.

Supp'- Figure 1. Specificity to IL-8 (increased levels) is surprising given one may expect a more global change in inflammatory status in response to MV induced trauma. Please comment

We were also surprised by these findings but find it interesting that the response to in vitro barotrauma in primary human macrophages specifically upregulates IL8. Because we used primary human alveolar macrophages from multiple donors there was a significant amount of variability in the induction of other cytokines such as IL6 and TGFbeta following barotrauma (Supplemental Figure 1) that limited our ability to find statistically significant differences in these cytokines.

Figure 1. "To determine if force induced miR-146a is observed clinically, miR-146a was measured in BAL cells obtained from a cohort of patients that underwent a clinically indicated bronchoscopy" What is the cellular make-up of the BALF- e.g. neutrophil to macrophage numbers?

The differential cell counts are included in Table 1. The BAL fluid contains a mixed inflammatory infiltrate consisting primarily of macrophages and neutrophils. However, as described in our response to Reviewer 1 (see comment #2), we have conducted new studies which demonstrate

that neutrophils do not consistently upregulate miR-146a levels to the same degree as macrophages. A summary of the data has been added to the Results and Discussion sections and are shown in Supplemental Figure 2.

Figure 2. In my opinion a 10-fold increase in the levels of a miR is more than a modest increase in the airways under such conditions. This is a very important correlative figure implicating MiR-146a to physiological function.

We agree with the reviewer's comment and have removed the word "modest" in this section as suggested.

Figure 3. This data shows an important protective role for miR146a in the injury response to ventilation. However, what happens after the acute phase (4 hours)? Especially to SPO2 and elastance? Can the authors provide data or comment on the development of lesion and pathophysiology in the more chronic phase?

We agree that it would be very interesting to examine what happens after the acute phase of injury. Unfortunately, our current animal model only allows for terminal experiments that last for several hours. The chronic phase of lung injury in patients with ARDS that receive mechanical ventilation is characterized by fibrosis that takes days to weeks to develop.(9, 10) The longest study of mechanical ventilation in mice that we are aware of was published in 2019 by Szabari et al who ventilated mice for 16 hours.(11) In our laboratory we have ventilated mice for up to 8hrs (3) and have observed that changes in lung compliance and oxygenation between different groups of mice typically become more pronounced by around 4 hours. Another related and interesting question would be to interrogate the effect of miR-146a overexpression on recovery from lung injury after the cessation of mechanical ventilation. Unlike in patients whom are removed from mechanical ventilation frequently, discontinuing mechanical ventilation in mice and allowing them to recover is technically challenging and we are not currently aware of any groups that routinely do this. In the future, we hope to address this technical limitation and develop a model that allows for recovery of the animals after mechanical ventilation.

Fig 4 and 5. The data would be strengthened if you could show a change in activation status (proinflammatory response) of the KO macrophages before and/or after transfer and in the specific uptake of miR-146a by macrophages after transfection.

As suggested, we generated new data examining the uptake of miR-146a by macrophages following lipid nanoparticle transfection. Cellular uptake was examined using Cy3 labeled lipid nanoparticles (LNPs) delivered via the intratracheal route followed by immunofluorescence imaging (Figure 6A-B and Supplemental Figure 7). Co-localization analysis indicated that ~44% of the nanoparticles were delivered to epithelial cells while ~52% were delivered to alveolar macrophages (Figure 6C). The percentage of macrophage area that contained nanoparticles was significantly higher than the percentage of epithelial cell area that contained nanoparticles (Figure 6D). Unfortunately, we were not able to examine the activation status of the miR-146a KO BMDMs in Figure 4 because all of the cells were used to generate the data in the experiment. We appreciate the suggestion and plan to look at activation status in future experiments.

Fig. 5. lipofectamine or nanoparticle (i.e. cationic lipoplex) based transfection techniques were used to overexpress miR-146a by 100-fold in primary human AMs (Figure 5A). – this is increasing the amount of miR-146a in these cells NOT increasing the expression.

We have edited the manuscript to incorporate this suggestion.

Fig. 5. And 6. What is the percentage of transfection of macrophages in the lung of mice? Are they the major cell where miR146 levels are increased?

As described in the Results and shown in the new Figure 6A-D and Supplemental Figure 7, we conducted additional experiments where lipid nanoparticles were loaded with the control pre-miR conjugated to a Cy3 fluorescent dye and delivered to mice via intratracheal injection. Immunofluorescence imaging and colocalization analysis was then used to determine distribution of cellular uptake. Co-localization analysis (Figure 6 C-D) indicated that ~44% of the nanoparticles were taken up by epithelial cells while ~52% were taken up by alveolar macrophages. However, given the significantly lower cell area of alveolar macrophages relative to epithelial cells, the percentage of AMs with nanoparticle uptake was significantly higher (~30%) compared the percentage of epithelial cells with nanoparticle uptake (~1%).

Reviewer #3 (Remarks to the Author):

This is an interesting paper with a focus on the biology of miR-146a and its role in lung macrophages mechanotransduction and lung injury. Overexpression of miR-146a has clear effects in the studies presented, but it is not clear how relevant this potential treatment would be for clinical use. There is also a tight focus on alveolar macrophages (AMs), but many other cell types may be affected by miR-146a overexpression in vivo, not to mention the various pathways that miRNAs in general can hit which further complicate the studies. It would also be nice to see an animal model with a clear readout for efficacy; something to show that this therapy makes a meaningful impact on animal health or survival, as well as to put the therapy in context with a clear clinical unmet need. It seems that this paper is highly relevant for a specialized readership, but not of sufficiently broad interest for a multidisciplinary journal due to the tight focus on miR-146a and the modest demonstration of therapeutic utility with a nanoparticle that is not likely to be useful clinically.

We thank the reviewer for these comments. The current standard of care for managing patients with ARDS is to use mechanical ventilation for life-support and to treat the inciting cause of ARDS (e.g. sepsis, trauma, COVID19) with the hope that lung injury will resolve. Unfortunately, clinicians in the ICU do not have pharmacologic therapies targeting the molecular pathophysiology of ARDS and there is a dire need for therapies that can treat or prevent the development of lung injury during ARDS. Our data demonstrate that modulating miR-146a levels may be a viable strategy for mitigating lung injury in the context of ARDS. We have added additional text to the Introduction section regarding the unmet clinical need in this field.

We also agree with the reviewer that showing a meaningful impact on animal health is important. The physiologic changes in lung function and oxygenation that we see in our model are important functional readouts and demonstrate that lung function is improved when miR-146a levels are increased. Many studies in the lung injury field do not include these functional parameters and instead rely on other indices of lung injury such as inflammatory markers and

measurements of barrier dysfunction. Although mortality is a very relevant endpoint in human studies, the overwhelming majority of preclinical models that use mechanical ventilation do not cause injury that is severe enough to result in animal mortality.(12-15) One recently published study did show increased mortality with prolonged mechanical ventilation,(11) but the cause of mortality has been the subject of significant debate and some experts feel that it is unlikely that death was due to lung injury in this model for several reasons. 1) The authors show an increase in mortality with VILI at early timepoints that our lab and many others use and very rarely have animals die. Our lab routinely ventilates mice for up to 8hrs with 0% mortality and this group reports a mortality rate of 40% at this timepoint. 2) The partial pressure of oxygen (PaO₂) in the blood of the mice in this study was >80 mm Hg which corresponds to an oxygen saturation >90% and would not result in death from hypoxemia or lung dysfunction. 3) All of the animals in the previously published study had a low pH from metabolic acidosis with high lactate levels likely due to shock/hypotension which may result from inadequate resuscitation or anesthetic dosing. These data suggest that the mice in the previously published study are dying from hemodynamic collapse and not lung dysfunction. Therefore, there is currently no reliable animal model of mortality due to VILI, and we feel that our physiological endpoints (lung function and oxygenation) provide the best evidence that miR-146a overexpression can reduce VILI. Additionally, our findings using nanoparticle delivery of a microRNA have implications for other forms of lung injury and in other types of organ injury.

1. How can one be sure the effects (Figure 6) are due to AMs and not other cells?

We cannot be sure the effects are completely specific to AMs at this point but our newly generated data in Supplemental Figure 2 demonstrate that the endogenous increase in miR-146a levels in AMs is nearly 100 fold higher than that seen in neutrophils which are the other inflammatory cells typically found in the lung during injury. Furthermore, we conducted new experiments where lipid nanoparticles were loaded with a control pre-miR conjugated to a fluorescent dye (i.e. Cy3) and delivered these nanoparticles to mice via intratracheal injection. Immunofluorescence imaging was then used to determine distribution and cellular uptake. Interestingly, our co-localization analysis (Figure 6 C&D) demonstrates that ~44% of the nanoparticles are delivered to epithelial cells while ~52% are delivered to alveolar macrophages. However, given the significantly lower cell area of alveolar macrophages relative to epithelial cells, the percentage of AMs with nanoparticle uptake was significantly higher (~30%) compared the percentage of epithelial cells with nanoparticle uptake (~1%). These data indicate that the mitigation of lung injury seen in Figure 6 is due in large part to uptake of miR-146a loaded lipid nanoparticles by AMs.

2. In addition to TRAF6, miR-146a likely silencing additional genes. One should look at gene signatures following miR-146a delivery more broadly to understand globally what all targets are inhibited. Some of these may be having a larger effect than TRAF6 or there could be combinatorial effects. There are many papers on miRNAs showing broad effects on gene expression broken into various pathways and cellular programs (bioinformatic analysis, clustering, etc.). This is likely relevant to the effects seen and lung injury treatment and should be studied.

We thank the reviewer for this helpful suggestion. We measured expression of other known miR-146a targets including IRAK-1 and SMAD-4 in addition to TRAF6. We did see a significant

decrease in SMAD4 levels in BAL cells following miR-146a nanoparticle delivery. Interestingly we did not see a difference in IRAK-1 levels in BAL cells. Not surprisingly we did not see any differences in message levels for any of these targets in whole lung homogenates which contains a mixture of various cells types many of which do not take up nanoparticles based on our new data in Figure 6A-D. We have added these data to the new Supplemental Figure 8. Although it would be interesting to perform gene expression profiling in our models to discover potential novel targets, we do not currently have the resources to perform these experiments. We are planning to include these types of experiments in an upcoming grant submission and hope to pursue this line of investigation in the future.

3. There is very little discussion of the delivery vehicle details in the main text (chemistry used, specific characteristics, biophysical properties, etc.). The experimental section details a procedure whereby miRNA was complexed with PEI2000, which would initially make uncontrolled polyplexes of various diameters. Separately, “empty liposomes” were made by combining DOPE, DOTAP, linoleic acid, and TPGS in HEPES buffer. Given that these molecules are insoluble in water (HEPES buffer), it is confusing how this process could be conducted. Were these water insoluble lipids sonicated to form liposomes or lipid nanoparticles (LNPs)? Were they actually prepared by dissolving in an organic, water miscible solvent like ethanol, to then combine with HEPES buffer (the “ethanol dilution method”)? Nevertheless, it is unclear what was actually made. Imaging by electron microscopy (e.g. TEM) could help tell if one made liposomes (with a bilayer) or LNPs (with a solid core). This is critical because they next step involves “loading” PEI/miRNA lipoplexes inside of presumably empty liposomes simply by incubating. It is also unclear how this would happen. One could imagine electrostatic interactions between anionic miRNAs and cationic materials (PEI, DOTAP, etc.), but what would be a driving force to load polyplexes inside of liposomes? Could liposomes / lipoplex complexes have been made instead? More details and characterization are needed.

We thank the reviewer for this comment and we agree that our original submission did not adequately describe the nanoparticle fabrication process or provide enough characterization information to determine exactly what type of nanoparticle was made. Therefore, we significantly revised the methods and conducted several additional characterization experiments. First, we now note in the Methods section that empty lipid nanoparticles were fabricated using the ethanol dilution method where DOPE, DOTAP, linoleic acid and TPGS were diluted in ethanol in a molar ratio of 40:10:48:2 and this mixture was injected into 20mM HEPES buffer. We also note that this solution was sonicated to form empty lipid nanoparticles. Pre-miR-146a was mixed with PEI in HEPES buffer and incubated at room temperature for 10 minutes to form PEI/miRNA polyplexes. The empty lipid nanoparticle solution and the polyplex solution was then mixed and sonicated for 5 min followed by 10 min incubation at room temperature. As shown in the new Supplemental Figure 6A, cryo-TEM images indicate that this procedure produces lipid nanoparticles (LNPs) with a solid core. In addition, to determine encapsulation efficiency (i.e. quantity of encapsulated miR-146a vs free miR-146a), we used agarose gel electrophoresis. As shown in Supplemental Figure 6D, when the solution containing miR-146a loaded LNP was treated with SDS to disrupt the LNPs, we detected significant amounts of miR-146a (blue box). However, when the miR-146a loaded LNPs were not treated with SDS we detected very low (if any) miR-146a (red box). This indicates that nearly 100% of the miR is encapsulated within the LNP.

4. There is a discussion about how size can affect pulmonary delivery, and a sentence that says the particles used had an average diameter of 123nm. Is this a suitable size? What about PDI / uniformity, miRNA encapsulation, dose? 77% encapsulation efficiency is reported in the SI, but how much was attempted to be loaded? There is a lack of details in the main text about the delivery experiment, readouts, etc.

Thank you for this comment and we agree a better discussion of how particle size influences pulmonary delivery is warranted. It is well established that aerosols with a 1-5um aerodynamic diameter exhibit optimal delivery to the distal lung.(16) Aerosols >5um impact on upper airways and are rapidly cleared while aerosols <500nm can be exhaled before deposition. However, particles <500nm have superior deposition on alveolar walls due to enhanced diffusion properties and smaller particles (~100-200nm) are known to facilitate cellular uptake.(17) Importantly, in this study, a liquid solution of miR-loaded LNPs was intratracheally instilled and not “aerosolized.” Therefore, the use of ~150nm particles is suitable for this study since the removal of aerosolized particles is not a concern. However, we acknowledge that clinical translation may require conversion of nanoparticle suspensions into inhalable droplets with 1–5 um aerodynamic diameters and that this has been performed by several previous investigators.(18)

Regarding uniformity, we measured the polydispersity index (PDI) of the nanoparticles (0.393 ± 0.022) and now report this in the manuscript. We performed agarose gel electrophoresis to measure encapsulation efficiency in vitro and found nearly 100% encapsulation of miR-146a into the nanoparticles (Supplemental Figure 6D). For in vivo experiments 1 nmol of miR was administered to mice. The Methods and Results sections have been significantly edited and now incorporate these details.

5. It is unclear how much miRNAs was packaged and delivered in vivo. It is also unclear how much miRNA reached targeted cells following intratracheal installation. Some characterization of dosage is needed. And some evidence that the miRNAs are actually in the AMs is needed as well. One could do this by fluorescence, radiolabeling, or other methods to track miRNA and/or nanoparticle uptake in lung cells in vivo.

For packaging and delivery, as described in the Methods, miR-146a loaded lipid nanoparticles were concentrated to 1nmol miR in 50uL by spinning at 3500g and then 50uL of miR-loaded LNPs were intratracheally instilled into mice 24 hours and 0 hours before ventilation.

Regarding evidence that the miRNAs are actually in AMs, we conducted immunofluorescence studies as suggested by the reviewer. Please refer comment 1 above, Figure 6A-D and Supplemental Figure 7 for details. Briefly, these studies demonstrated that ~30% uptake in AMs with only ~1% uptake in epithelial cells based on colocalization analysis.

1. Goligher EC, Ferguson ND, and Brochard LJ. Clinical challenges in mechanical ventilation. *Lancet*. 2016;387(10030):1856-66.
2. Slutsky AS, and Ranieri VM. Ventilator-induced lung injury. *N Engl J Med*. 2013;369(22):2126-36.
3. Boudreault F, Pinilla-Vera M, Englert JA, Kho AT, Isabelle C, Arciniegas AJ, et al. Zinc deficiency primes the lung for ventilator-induced injury. *JCI Insight*. 2017;2(11).

4. Hegeman MA, Hemmes SN, Kuipers MT, Bos LD, Jongasma G, Roelofs JJ, et al. The extent of ventilator-induced lung injury in mice partly depends on duration of mechanical ventilation. *Crit Care Res Pract.* 2013;2013:435236.
5. Vaneker M, Halbertsma FJ, van Egmond J, Netea MG, Dijkman HB, Snijdelaar DG, et al. Mechanical ventilation in healthy mice induces reversible pulmonary and systemic cytokine elevation with preserved alveolar integrity: an in vivo model using clinical relevant ventilation settings. *Anesthesiology.* 2007;107(3):419-26.
6. Belperio JA, Keane MP, Burdick MD, Londhe V, Xue YY, Li K, et al. Critical role for CXCR2 and CXCR2 ligands during the pathogenesis of ventilator-induced lung injury. *J Clin Invest.* 2002;110(11):1703-16.
7. Hoetzel A, Schmidt R, Vallbracht S, Goebel U, Dolinay T, Kim HP, et al. Carbon monoxide prevents ventilator-induced lung injury via caveolin-1. *Crit Care Med.* 2009;37(5):1708-15.
8. Yu Z, Wang T, Zhang L, Yang X, Li Q, and Ding X. WISP1 and TLR4 on Macrophages Contribute to Ventilator-Induced Lung Injury. *Inflammation.* 2020;43(2):425-32.
9. Cabrera-Benitez NE, Laffey JG, Parotto M, Spieth PM, Villar J, Zhang H, et al. Mechanical ventilation-associated lung fibrosis in acute respiratory distress syndrome: a significant contributor to poor outcome. *Anesthesiology.* 2014;121(1):189-98.
10. Thompson BT, Chambers RC, and Liu KD. Acute Respiratory Distress Syndrome. *N Engl J Med.* 2017;377(6):562-72.
11. Szabari MV, Takahashi K, Feng Y, Locascio JJ, Chao W, Carter EA, et al. Relation between Respiratory Mechanics, Inflammation, and Survival in Experimental Mechanical Ventilation. *Am J Respir Cell Mol Biol.* 2019;60(2):179-88.
12. Lopez-Alonso I, Blazquez-Prieto J, Amado-Rodriguez L, Gonzalez-Lopez A, Astudillo A, Sanchez M, et al. Preventing loss of mechanosensation by the nuclear membranes of alveolar cells reduces lung injury in mice during mechanical ventilation. *Sci Transl Med.* 2018;10(456).
13. Makena PS, Gorantla VK, Ghosh MC, Bezawada L, Kandasamy K, Balazs L, et al. Deletion of apoptosis signal-regulating kinase-1 prevents ventilator-induced lung injury in mice. *Am J Respir Cell Mol Biol.* 2012;46(4):461-9.
14. Siempos, II, Ma KC, Imamura M, Baron RM, Fredenburgh LE, Huh JW, et al. RIPK3 mediates pathogenesis of experimental ventilator-induced lung injury. *JCI Insight.* 2018;3(9).
15. Wienhold SM, Macri M, Nouailles G, Dietert K, Gurtner C, Gruber AD, et al. Ventilator-induced lung injury is aggravated by antibiotic mediated microbiota depletion in mice. *Crit Care.* 2018;22(1):282.
16. Dhand C, Prabhakaran MP, Beuerman RW, Lakshminarayanan R, Dwivedi N, and Ramakrishna S. Role of size of drug delivery carriers for pulmonary and intravenous administration with emphasis on cancer therapeutics and lung-targeted drug delivery. *RSC Adv.* 2014;4(62):32673-89.
17. El-Sherbiny IM, El-Baz NM, and Yacoub MH. Inhaled nano- and microparticles for drug delivery. *Glob Cardiol Sci Pract.* 2015;2015:2.
18. Mangal S, Gao W, Li T, and Zhou QT. Pulmonary delivery of nanoparticle chemotherapy for the treatment of lung cancers: challenges and opportunities. *Acta Pharmacol Sin.* 2017;38(6):782-97.

REVIEWERS' COMMENTS:

Reviewer #2 (Remarks to the Author):

The authors have adequately addresses all of my concerns

Reviewer #3 (Remarks to the Author):

Although I still have some reservations about this manuscript, I greatly appreciate the attention to detail and additional experiments performed to address nearly all reviewer comments. As such, I recommend to accept if all other reviewers and the editor agree.

A few remaining minor comments:

Regarding the miRNA encapsulation data, it is improper to assume "nearly 100%" encapsulation (main text, page 16, line 269) without measuring it quantitatively. I agree that the gel shows encapsulation (Figure S6d), but without quantification of the gel image, one cannot assert a number. A better method would be an assay such as the Quant-iT RiboGreen RNA Assay (Thermo Fisher), which is widely used to quantify miRNA encapsulation. I suggest to remove this number or, preferably, to just quantify it using RiboGreen.

Regarding Figure 6A-D, I suggest to be careful about how the data is described in the main text and plotted. I definitely appreciate this addition, as it resolves a previous shortcoming. However, the images in Figure 6A show that a small number of cells are transfected overall. From the response letter and Figure 6D, 30% of AMs internalized nanoparticles and 1% of epithelial cells internalized nanoparticles. As such, the added Main Text is a little misleading, as it reads that "~44% of the nanoparticles were delivered to epithelial cells while ~52% were delivered to alveolar macrophages", which may be true from the basis of added nanoparticles and where they went, but this sentence is misleading in that a reader may interpret that sentence to mean that 44% of ECs and 52% of AMs were transfected (which they were not). So I suggest to significantly revise these sentences prior to publication to more accurately and fairly summarize the results.

Regarding the nanoparticle work, I suggest in future studies to directly encapsulate miRNAs into LNPs by the ethanol dilution method. The PEI polyplex formation and subsequent encapsulation into pre-formed LNPs is complicated and does not yield monodisperse LNPs (PDI = 0.4, which is broad as reflected in the Fig. 6b histogram showing particles up to 1 micrometer in diameter). The added PEI would not be necessary if an ionizable cationic lipid (e.g. containing a tertiary amine, pKa ~ 6.3) were used instead, which could be directly used in an LNP with miRNA encapsulated in situ by dissolving the miRNA in acidic buffer for mixing with lipids in ethanol. Given the stated goal to be clinically relevant, the authors might even consider using Alnylam's Onpattro formulation, which is FDA-approved for siRNA delivery to the liver. The FDA label reveals the details, and all lipid components (DLin-MC3-DMA, DSPC, cholesterol, PEG2000-DMG) are now commercially available to purchase and inexpensive. If not effective in lung cells, one could supplement with additional molecules or optimize further. This is not a suggestion to change anything about the current manuscript. The comment is simply for future experiments, that the LNP aspect of the research could be improved for future research projects to increase efficacy and even further improve the already exciting outcomes.

Reviewer 3 comments

Although I still have some reservations about this manuscript, I greatly appreciate the attention to detail and additional experiments performed to address nearly all reviewer comments. As such, I recommend to accept if all other reviewers and the editor agree.

A few remaining minor comments:

Regarding the miRNA encapsulation data, it is improper to assume "nearly 100%" encapsulation (main text, page 16, line 269) without measuring it quantitatively. I agree that the gel shows encapsulation (Figure S6d), but without quantification of the gel image, one cannot assert a number. A better method would be an assay such as the Quant-iT RiboGreen RNA Assay (Thermo Fisher), which is widely used to quantify miRNA encapsulation. I suggest to remove this number or, preferably, to just quantify it using RiboGreen.

We have removed the term "100%" as suggested by the reviewer.

Regarding Figure 6A-D, I suggest to be careful about how the data is described in the main text and plotted. I definitely appreciate this addition, as it resolves a previous shortcoming. However, the images in Figure 6A show that a small number of cells are transfected overall. From the response letter and Figure 6D, 30% of AMs internalized nanoparticles and 1% of epithelial cells internalized nanoparticles. As such, the added Main Text is a little misleading, as it reads that "~44% of the nanoparticles were delivered to epithelial cells while ~52% were delivered to alveolar macrophages", which may be true from the basis of added nanoparticles and where they went, but this sentence is misleading in that a reader may interpret that sentence to mean that 44% of ECs and 52% of AMs were transfected (which they were not). So I suggest to significantly revise these sentences prior to publication to more accurately and fairly summarize the results.

We have revised the text of the Results section for clarity as suggested by the reviewer.

Regarding the nanoparticle work, I suggest in future studies to directly encapsulate miRNAs into LNPs by the ethanol dilution method. The PEI polyplex formation and subsequent encapsulation into pre-formed LNPs is complicated and does not yield monodisperse LNPs (PDI = 0.4, which is broad as reflected in the Fig. 6b histogram showing particles up to 1 micrometer in diameter). The added PEI would not be necessary if an ionizable cationic lipid (e.g. containing a tertiary amine, pKa ~ 6.3) were used instead, which could be directly used in an LNP with miRNA encapsulated in situ by dissolving the miRNA in acidic buffer for mixing with lipids in ethanol. Given the stated goal to be clinically relevant, the authors might even consider using Alnylam's Onpattro formulation, which is FDA-approved for siRNA delivery to the liver. The FDA label reveals the details, and all lipid components (DLin-MC3-DMA, DSPC, cholesterol, PEG2000-DMG) are now commercially available to purchase and inexpensive. If not effective in lung cells, one could supplement with additional molecules or optimize further. This is not a suggestion to change anything about the current manuscript. The

comment is simply for future experiments, that the LNP aspect of the research could be improved for future research projects to increase efficacy and even further improve the already exciting outcomes.

We thank the reviewer for this helpful suggestion and will explore this option for future experiments.